# Towards Unified and Lossless Latent Space for 3D Molecular Latent Diffusion Modeling

**Yanchen Luo[1], Zhiyuan Liu[2]\*, Yi Zhao[1], Sihang Li[1], Hengxing Cai[3],**
**Kenji Kawaguchi[2], Tat-Seng Chua[2], Yang Zhang[2], Xiang Wang[1]\***
[1]University of Science and Technology of China
[2]National University of Singapore
[3]DP Technology
`luoyanchen@mail.ustc.edu.cn`
`acharkq@gmail.com`
`xiangwang1223@gmail.com`

## Abstract

3D molecule generation is crucial for drug discovery and material science, requiring models to process complex multi-modalities, including atom types, chemical bonds, and 3D coordinates. A key challenge is integrating these modalities of different shapes while maintaining SE(3) equivariance for 3D coordinates. To achieve this, existing approaches typically maintain separate latent spaces for invariant and equivariant modalities, reducing efficiency in both training and sampling. In this work, we propose **U**nified Variational **A**uto-**E**ncoder for **3D** Molecular Latent Diffusion Modeling (**UAE-3D**), a multi-modal VAE that compresses 3D molecules into latent sequences from a unified latent space, while maintaining near-zero reconstruction error. This unified latent space eliminates the complexities of handling multi-modality and equivariance when performing latent diffusion modeling. We demonstrate this by employing the Diffusion Transformer–a general-purpose diffusion model without any molecular inductive bias–for latent generation. Extensive experiments on GEOM-Drugs and QM9 datasets demonstrate that our method significantly establishes new benchmarks in both *de novo* and conditional 3D molecule generation, achieving leading efficiency and quality. On GEOM-Drugs, it reduces FCD by 72.6% over the previous best result, while achieving over 70% relative average improvements in geometric fidelity. Our code is released at `https://github.com/lyc0930/UAE-3D/`.

## 1 Introduction

The discovery of novel molecules is fundamental to various scientific fields, particularly in drug and material development. Given significant progress has been made in designing 2D molecular graphs [1–4], recent research has increasingly focused on the generation of 3D molecules [5, 6]. Unlike 2D molecular generation, which focuses on forming valid molecular structures based on chemical bonds, 3D generation must also predict 3D atomic coordinates that align with the 2D structure. Accurate 3D molecule generation is essential to power many important applications, such as structure-based drug design [7] and inverse molecule design targeting quantum properties [8].

3D molecule generation is challenging due to its multi-modal nature. As shown in Figure 1(a), a 3D molecule consists of features of three distinct modalities: atom types, atomic coordinates, and chemical bonds. This requires the generation model to handle both discrete (*e.g.,* atom types) and continuous features (*e.g.,* coordinates), while also addressing differences in feature shapes (atom-wise

---

\*Corresponding authors.

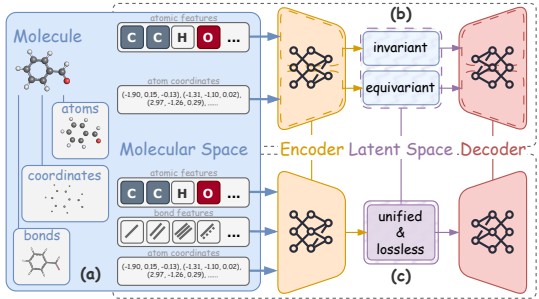

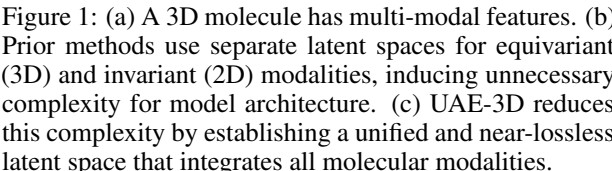

Figure 1: (a) A 3D molecule has multi-modal features. (b) Prior methods use separate latent spaces for equivariant (3D) and invariant (2D) modalities, inducing unnecessary complexity for model architecture. (c) UAE-3D reduces this complexity by establishing a unified and near-lossless latent space that integrates all molecular modalities.

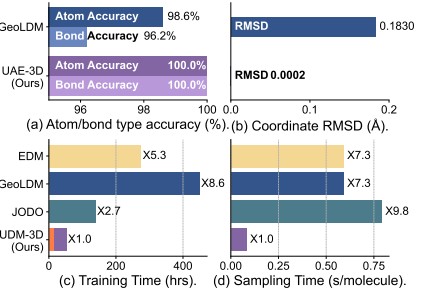

Figure 2: Comparing UAE-3D and UDM-3D with other methods on the QM9 dataset. (a;b) Reconstruction errors on the test set. (c;d) Comparing training and inference time.

*v.s.* edge-wise features). Worse still, the modality of 3D coordinate requires special care to respect rotational and translational equivariance, whereas other modalities do not, further complicating the task. Mitigating this multi-modal challenge, prior works mostly process each modality in separate latent spaces: some maintain separate diffusion processes for each modality [6, 9–11], while others interleave the prediction of different modalities across autoregressive generation steps [12, 13]. However, we argue that handling each modality separately complicates the model design, reducing both the training and sampling efficiency. Moreover, processing each modality separately risks compromising the consistency between them. These issues raise a critical question: *Can we design a unified generative model that seamlessly integrates all three modalities of 3D molecule generation?*

To answer this research question, we propose to build a multi-modal latent diffusion model (LDM) [14] for the unified generative modeling of 3D molecules. LDM extends the diffusion paradigm [15] by operating in a compressed latent space learned through variational autoencoders (VAEs) [16], offering improved computational efficiency and generation quality. For 3D molecules, we can build a multi-modal VAE that compresses all three modalities in a single unified latent space. Scrutinizing the previous works, we identify the following challenges to achieve this purpose:

- **The Challenge of Compressing Multi-modality in One Latent Space.** While previous studies have explored LDMs for 3D molecule and protein generation [17, 18], they fail to build a unified latent space that integrates all modalities. Their difficulty arises from the reliance on neural networks with baked-in 3D equivariance [19, 6], which mostly maintain separate latent spaces for 3D equivariant and invariant modalities (*cf.* Figure 1(b)).

- **The Challenge of Ensuring Near-Lossless Compression.** Ensuring a low reconstruction error for the molecular VAE is critical, as even small errors can result in invalid or unstable 3D structures. Furthermore, imprecise latents that cannot reproduce the original molecule can propagate their errors to the LDM, easily disrupting the generative modeling. Despite its critical importance, reconstruction error has been largely overlooked in previous works [17, 18] (*cf.* Figure 2(a;b)).

In this work, we introduce **UAE-3D**, **U**nified Variational **A**uto-**E**ncoder for **3D** Molecular Latent Diffusion Modeling, a VAE that can compress the multi-modal features of 3D molecules into a unified latent space while maintaining near zero (100% atom/bond accuracy and 2E-4 coordinate RMSD) reconstruction error. To obtain a latent space integrating both 3D equivariant and invariant modalities, we draw inspiration from the "bitter lesson" [20] of 3D molecules: rather than using models with intricate, baked-in 3D equivariance, UAE-3D trains a neural network to "learn" 3D equivariance through our tailor-made SE(3) augmentations, encouraging the transformation on the input coordinates to be reflected equivariantly on the output coordinates. Moreover, UAE-3D employs the Relational Transformer [21] as its encoder, leveraging its scalability and flexibility to incorporate both atom-wise and edge-wise features. A Transformer-based [22] decoder is jointly trained with the encoder to reconstruct both 3D invariant and equivariant molecular features.

Despite its simplicity, UAE-3D can compress 3D molecules into token sequences in a unified latent space, **eliminating the complexities of handling multi-modalities and 3D equivariance in latent diffusion modeling.** Figure 2(a;b) show that UAE-3D reaches near-lossless reconstruction of atom and bond types, with a near-zero coordinate RMSD. To further demonstrate its effectiveness, we

employ the Diffusion Transformer (DiT) [23], a general-purpose diffusion backbone without any molecular inductive bias, to model UAE-3D's latents. We show that DiT can successfully generate stable and valid 3D molecules, with significantly improved training (by 2.7 times) and sampling (by 7.3 times) efficiency (*cf.* Figure 2(c;d)). We refer to this LDM pipeline as **UDM-3D**: Unified Latent Diffusion Modeling for **3D** Molecule Generation.

Extensive experiments demonstrate that UDM-3D, powered by UAE-3D, achieves state-of-the-art results in both *de novo* and conditional 3D molecule generation on the QM9 [24] and GEOM-Drugs [25] datasets. On GEOM-Drugs, our model achieves a 72.6% reduction in Fréchet ChemNet Distance [3], while significantly reducing the MMD of bond length, bond angle, and dihedral angle by 88.4%, 55.6%, and 74.0%, respectively. In conditional generation on QM9, it achieves the lowest MAE in five out of six properties. These results confirm the effectiveness of our unified modeling approach. Ablation studies show the effectiveness of our key components. Further analysis reveals that UAE-3D's latents present structured variations across geometric movement.

## 2 Background: 3D Molecular Latent Diffusion Models

A 3D molecular LDM involves a 3D molecular VAE to compress 3D molecules into the latent space, where a diffusion model performs generative modeling. Below, we introduce both components.

**Notations.** A 3D molecule is represented by $\mathbf{G} = \langle \mathbf{F}, \mathbf{E}, \mathbf{X} \rangle$, where $\mathbf{F} \in \mathbb{R}^{|\mathcal{V}| \times d_1}$ is the atom feature matrix (*e.g.,* atomic numbers), $\mathbf{E} \in \mathbb{R}^{|\mathcal{V}| \times |\mathcal{V}| \times d_2}$ is the bond feature matrix (*e.g.,* bond connectivity and type), and $\mathbf{X} \in \mathbb{R}^{|\mathcal{V}| \times 3}$ is the 3D atom coordinate matrix. $d_1$ and $d_2$ are dimensions of atom and bond features. $\mathcal{V}$ is $\mathbf{G}'s$ set of atoms and $|\mathcal{V}|$ denotes the number of atoms.

**3D Molecular Variational Auto-Encoding.** A molecular VAE consists of an encoder $\mathcal{E}$ and a decoder $\mathcal{D}$. Given a 3D molecule $\mathbf{G}$, the encoder $\mathcal{E}$ maps it into a sequence of latent tokens $\mathcal{E}(\mathbf{G}) = \mathbf{Z} = \{\boldsymbol{z}_i \in \mathbb{R}^d | i \in \mathcal{V}\}$. Each latent $\boldsymbol{z}_i$ is sampled from a Gaussian distribution $\mathcal{N}(\boldsymbol{z}_i; \boldsymbol{\mu}_i, \boldsymbol{\sigma}_i)$ using the reparameterization trick [16], where $\boldsymbol{\mu}_i$ and $\boldsymbol{\sigma}_i$ are learned parameters for atom $i$. The decoder $\mathcal{D}$ then reconstructs $\mathbf{G}$ from these latents: $\hat{\mathbf{G}} = \mathcal{D}(\mathbf{Z})$. The VAE is trained with a reconstruction loss and a regularization term:

$$\mathcal{L}_{\text{VAE}} = \mathbb{E}_{\mathbf{G} \sim p_{\text{data}}} \left[ \|\hat{\mathbf{G}} - \mathbf{G}\| + D_{\text{KL}}(q(\mathbf{Z}|\mathbf{G})\|p(\boldsymbol{z})) \right], \tag{1}$$

where $D_{\text{KL}}$ denotes the Kullback-Leibler divergence; $q(\mathbf{Z}|\mathbf{G}) = \prod_{i \in \mathcal{V}} \mathcal{N}(\boldsymbol{z}_i; \boldsymbol{\mu}_i, \boldsymbol{\sigma}_i)$ is the approximated posterior distribution; and $\|\hat{\mathbf{G}} - \mathbf{G}\|$ measures the difference between the original and the predicted graph, whose definition varies across different works. In [17], $\|\mathbf{G} - \hat{\mathbf{G}}\|$ combines an MSE loss for 3D coordinates and a cross-entropy loss for atom types.

**Diffusion Model.** Building on the molecular VAE's latent $\mathbf{Z}$, a diffusion model [14, 26] performs generative modeling. In forward diffusion process, we gradually add noise to the original latent $\mathbf{Z}^{(0)} = \mathbf{Z}$ following $\mathbf{Z}^{(t)} \sim \mathcal{N}(\mathbf{Z}^{(t)}; \sqrt{\bar{\alpha}^{(t)}}\mathbf{Z}^{(0)}, (1 - \bar{\alpha}^{(t)})\mathbf{I})$, where $t \in (0, 1]$ is the diffusion timestep, and $\bar{\alpha}^{(t)}$ is a hyperparameter controlling the signal/noise ratio. In practice, $\mathbf{Z}^{(t)}$ is sampled as $\mathbf{Z}^{(t)} = \sqrt{\bar{\alpha}^{(t)}}\mathbf{Z}^{(0)} + \sqrt{1 - \bar{\alpha}^{(t)}}\boldsymbol{\epsilon}$, where $\boldsymbol{\epsilon} \sim \mathcal{N}(\mathbf{0}, \mathbf{I})$. Given the noised latents $\mathbf{Z}^{(t)}$, a diffusion model $\boldsymbol{\epsilon}_\theta(\mathbf{Z}^{(t)}, t)$ is trained to predict the added noise $\boldsymbol{\epsilon}$ by minimizing the MSE loss $\|\boldsymbol{\epsilon}_\theta(\mathbf{Z}^{(t)}, t) - \boldsymbol{\epsilon}\|^2$. Once trained, new latents $\hat{\mathbf{Z}}^{(0)}$ are sampled with the model $\boldsymbol{\epsilon}_\theta$ via iterative denoising [15], and the decoder $\mathcal{D}$ reconstructs the corresponding 3D molecule via $\hat{\mathbf{G}} = \mathcal{D}(\hat{\mathbf{Z}}^{(0)})$.

**Separated Latent Spaces.** The VAE latents in previous works [18, 17] include two parts: $\mathbf{Z} = [\vec{\mathbf{Z}}; \bar{\mathbf{Z}}]$, where $\vec{\mathbf{Z}}$ is for equivariant features and $\bar{\mathbf{Z}}$ is for 3D invariant features. They also rely on diffusion models with baked-in 3D equivariance. In contrast, as Section 3 shows, our method uses a unified latent space, allowing a general-purpose diffusion model–without any geometric or molecular inductive bias–to achieve strong performance.

## 3 Methodology

In this section, we propose the **UAE-3D**, a multi-modal variational auto-encoder designed to effectively compress the diverse modalities of 3D molecules into a unified latent space. Based on UAE-3D, we introduce **UDM-3D**, an LDM for 3D molecule generation.

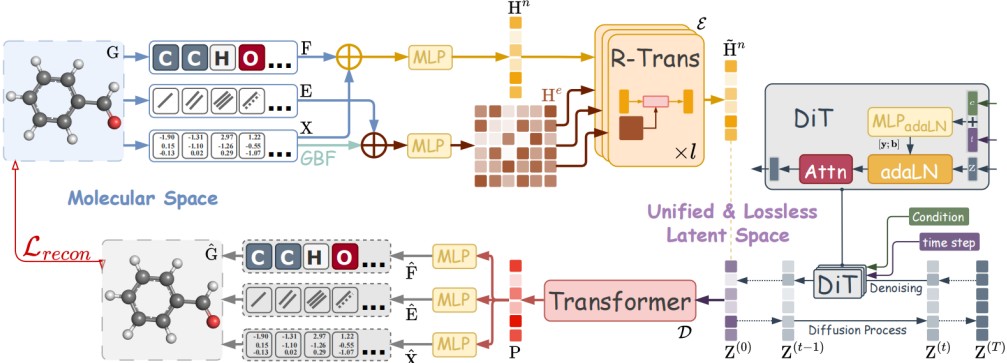

Figure 3: Overview of the UDM-3D and UAE-3D models. The UAE-3D encodes 3D molecules from molecular space into a unified latent space, integrating multi-modal features such as atom types, chemical bonds, and 3D coordinates. Utilizing this latent space, UDM-3D employs a DiT to perform generative modeling. Then, the denoised latents are decoded back into 3D molecules.

### 3.1 Unified Variational Auto-Encoder for 3D Molecular Latent Diffusion Modeling

UAE-3D is designed to address the complexities of multi-modal and equivariance of molecular data by compressing the atomic features, bond features, and atomic coordinates into a unified latent space. This is achieved through three key components: (1) a Relatinoal-Transformer [21] that effectively integrates the multi-modal features into token sequences, (2) a Transformer-based [22] decoder and the reconstruction loss for 3D molecule reconstruction; and (3) our tailor-made SE(3)-equivariant data augmentations to train the model to learn 3D equivariance.

**Compressing 3D Molecules with Relational Transformer (R-Trans) Encoder.** Given a 3D molecular $\mathbf{G} = \langle \mathbf{F}, \mathbf{E}, \mathbf{X} \rangle$, we compute its initial atom-wise and edge-wise embeddings as:

$$\mathbf{H}^n = \mathrm{MLP}([\mathbf{X}; \mathbf{F}]) \in \mathbb{R}^{|\mathcal{V}| \times d}, \quad (2) \qquad \mathbf{H}^e = \mathrm{MLP}([\mathbf{E}; \mathbf{D}]) \in \mathbb{R}^{|\mathcal{V}| \times |\mathcal{V}| \times d}, \quad (3)$$

where $\mathbf{D}_{ij} = \mathrm{GBF}(\|\mathbf{X}_i - \mathbf{X}_j\|^2) \in \mathbb{R}^d$; $\mathrm{GBF}(\cdot)$ implements the Gaussian basis functions to expand inter-atomic distances as feature vectors [27]; and $d$ is the embedding dimension. The initial node embeddings $\mathbf{H}^n$ combines both atomic features and its 3D coordinates, while the edge embeddings $\mathbf{H}^e$ incorporate bond features and inter-atomic distances. $\mathbf{H}^n$ and $\mathbf{H}^e$ together effectively represent all 3D molecular information for the subsequent encoding. Given these embeddings, we process them with the molecular encoder $\mathcal{E}(\cdot)$ of $L$ layers of R-Trans [21]:

$$\tilde{\mathbf{H}}^n = \mathrm{R\text{-}Trans}(\mathbf{H}^n, \mathbf{H}^e) \in \mathbb{R}^{|\mathcal{V}| \times d}, \tag{4}$$

where $\tilde{\mathbf{H}}^n$ denotes the updated atom embedding for the next layer, computed as follows:

$$\mathbf{Q}_{ij} = [\mathbf{H}^n_i; \mathbf{H}^e_{ij}]\mathbf{W}^q \in \mathbb{R}^d, \quad (5) \qquad [\mathbf{K}_{ij}; \mathbf{V}_{ij}] = [\mathbf{H}^n_j; \mathbf{H}^e_{ij}]\mathbf{W}^{kv} \in \mathbb{R}^{2d}, \quad (6)$$

$$\boldsymbol{\alpha}_{ij} = \mathrm{softmax}_j \left( \frac{\mathbf{Q}_{ij} \mathbf{K}_{ij}^\top}{\sqrt{d}} \right) \in \mathbb{R}, \quad (7) \qquad \hat{\mathbf{H}}^n_i = \mathrm{MLP}(\sum_j \boldsymbol{\alpha}_{ij} \mathbf{V}_{ij}) \in \mathbb{R}^d, \quad (8)$$

where $\mathbf{W}^q$ and $\mathbf{W}^{kv}$ are linear projectors. Unlike the original R-Trans, our implementation does not update the edge $\mathbf{H}^e$ every layer, which improves efficiency without compromising performance.

Compared to Transformer [22], R-Trans is well-suited for 3D molecule encoding, that it can effectively integrate edge embeddings $\mathbf{H}^e \in \mathbb{R}^{|\mathcal{V}| \times |\mathcal{V}| \times d}$ and atom embeddings $\mathbf{H}^n \in \mathbb{R}^{|\mathcal{V}| \times d}$, despite their different shapes. This integration occurs during the computation of queries, keys and values (Equation (5) and (6)), ensuring the output atom embeddings fully incorporate edge information.

**Decoder.** Given encoder's latents $\mathbf{Z}$, we employ a Transformer [22] decoder $\mathcal{D}(\cdot)$ to obtain the atom-wise output $\mathbf{P} = \mathrm{Transformer}(\mathbf{Z}) \in \mathbb{R}^{|\mathcal{V}| \times d}$. Unlike the encoder, our decoder includes no molecule-specialized design, because it processes sequences of latents. Further, we employ three MLP predictors for atom, bond types, and atom coordinates to reconstruct complete 3D molecules:

$$\hat{\mathbf{X}}_i = \mathrm{MLP}_1(\mathbf{P}_i) \in \mathbb{R}^3; \quad \hat{\mathbf{F}}_i = \mathrm{MLP}_2(\mathbf{P}_i) \in \mathbb{R}^{N_a}; \quad \hat{\mathbf{E}}_{ij} = \mathrm{MLP}_3(\mathbf{P}_i + \mathbf{P}_j) \in \mathbb{R}^{N_b}, \quad (9)$$

where $N_a$ and $N_b$ are the number of atom and bond types.

**Reconstruction Loss.** We define the distance between our reconstructed graph and the original graph $\|\hat{\mathbf{G}} - \mathbf{G}\|$. It includes multiple components to ensure the fidelity of generated 3D molecules:

$$\mathcal{L}_{\text{atom}} = \frac{1}{|\mathcal{V}|} \sum_{i \in \mathcal{V}} \text{CE}(\hat{\mathbf{F}}_i, \mathbf{F}_i), \qquad \mathcal{L}_{\text{coordinate}} = \frac{1}{|\mathcal{V}|} \sum_{i \in \mathcal{V}} \text{MSE}(\hat{\mathbf{X}}_i, \mathbf{X}_i),$$

$$\mathcal{L}_{\text{bond}} = \frac{1}{|\mathcal{V}|^2} \sum_{i,j \in \mathcal{V}} \text{CE}(\hat{\mathbf{E}}_{ij}, \mathbf{E}_{ij}), \qquad \mathcal{L}_{\text{distance}} = \frac{1}{|\mathcal{V}|^2} \sum_{i,j \in \mathcal{V}} w_{ij} \, \text{MSE}(\|\hat{\mathbf{X}}_i - \hat{\mathbf{X}}_j\|, \mathbf{D}_{ij}). \tag{10}$$

where CE denotes cross-entropy loss, and $\mathcal{L}_{\text{atom}}$ and $\mathcal{L}_{\text{bond}}$ are losses for atom and bond types; $\mathcal{L}_{\text{coordinate}}$ measures the MSE between the predicted and the ground truth coordinates; and $\mathcal{L}_{\text{distance}}$ serves as an extra constraint on coordinate predictions, to ensure correct inter-atomic distances. Notably, we incorporate a *bonded distance loss* inside $\mathcal{L}_{\text{distance}}$ by adjusting the pairwise weight $w_{ij}$:

$$w_{ij} = \begin{cases} 1 + \lambda, & \text{if atom } i \text{ and } j \text{ are bonded}, \\ 1, & \text{otherwise}. \end{cases} \tag{11}$$

This design prioritizes accurate inter-atomic distances between bonded atoms, enforcing stricter geometry in chemically important regions. $\lambda$ is a hyperparameter that controls this constraint's strength. Given the components above, we define the distance between our reconstructed graph and the original graph (*i.e.,* the reconstruction loss) as follows:

$$\mathcal{L}_{\text{recon}} = \mathbb{E}_{\mathbf{G} \sim p_{\text{data}}} \|\hat{\mathbf{G}} - \mathbf{G}\| = \mathbb{E}_{\mathbf{G} \sim p_{\text{data}}} \boldsymbol{\gamma} \cdot [\mathcal{L}_{\text{atom}}, \mathcal{L}_{\text{bond}}, \mathcal{L}_{\text{coordinate}}, \mathcal{L}_{\text{distance}}]^{\top}, \tag{12}$$

where $\boldsymbol{\gamma} \in \mathbb{R}^4$ is a hyperparameter vector balancing the reconstruction terms. The final loss for UAE-3D combines $\mathcal{L}_{\text{recon}}$ with a KL regularization term:

$$\mathcal{L}_{\text{UAE-3D}} = \mathcal{L}_{\text{recon}} + \beta \cdot \mathbb{E}_{\mathbf{G} \sim p_{\text{data}}} \left[ D_{\text{KL}}(q(\mathbf{Z}|\mathbf{G}) \| p(\mathbf{Z})) \right], \tag{13}$$

where the hyperparameter $\beta \in \mathbb{R}$ controls the KL regularization strength.

**SE(3)-Equivariant Augmentations.** To enforce SE(3)-equivariance in UAE-3D, we apply random transformations $\mathbf{R} \in \text{SE}(3)$ to input coordinates $\mathbf{X}$ during training [28, 20]. Each $\mathbf{R}$ combines a rotation from SO(3) and a random translation from $\mathcal{N}(\mathbf{0}, 0.01\mathbf{I}_3)$. Crucially, the reconstruction loss $\mathcal{L}_{\text{recon}}$ (*cf.* Equation (12)) remains unchanged but operates on transformed inputs $\mathbf{R} \circ \mathbf{G} = \langle \mathbf{F}, \mathbf{E}, \mathbf{R}(\mathbf{X}) \rangle$. This process can be informally conceptualized as a reconstruction loss of $\|\mathcal{D}(\mathcal{E}(\mathbf{R} \circ \mathbf{G})) - \mathbf{R} \circ \mathbf{G}\|$, where the autoencoder $\mathcal{D}(\mathcal{E}(\cdot))$ learns to preserve geometric consistency under SE(3) transformations.

## 3.2 Unified Latent Diffusion Modeling for 3D Molecule Generation

**Diffusion Transformer (DiT).** Given the unified latent space provided by our UAE-3D model, we adopt the DiT [23] as the backbone diffusion model $\epsilon_\theta$ for 3D molecular latent generation. Originally developed for image LDM, DiT has demonstrated strong performance in modeling latent sequences. Compared to a standard Transformer [22], DiT replaces the layernorm [29] by adaptive layernorm (adaLN) [30], where the scale $\mathbf{y} \in \mathbb{R}^d$ and shift $\mathbf{b} \in \mathbb{R}^d$ parameters are conditioned on the diffusion timestep $t \in (0, 1]$. Specifically, $[\mathbf{y}; \mathbf{b}] = \text{MLP}_{\text{adaLN}}(\text{Embed}_t(t))$, where $\text{Embed}_t(\cdot)$ is a shared linear layer and $\text{MLP}_{\text{adaLN}}$ is a module specific to each adaLN instance. These generated shift and scale parameters act as "soft gates", enabling timestep-dependent activation of DiT's hidden representations and facilitating more effective, scale-adaptive denoising.

**Conditional Generation with Classifier-Free Guidance.** To enable generation with a condition vector $\mathbf{c}$, we extend the adaLN modules by combining the condition embedding $\text{Embed}_c(\mathbf{c})$ with the timestep embedding $\text{Embed}_t(t)$ when computing the scale $\mathbf{y}$ and shift $\mathbf{b}$ parameters. This allows the diffusion model $\epsilon\theta(\mathbf{Z}^{(t)}, t, \mathbf{c})$ to incorporate conditioning information during denoising:

$$[\mathbf{y}; \mathbf{b}] = \text{MLP}_{\text{adaLN}}(\text{Embed}_t(t) + \text{Embed}_c(\mathbf{c})). \tag{14}$$

To further enforce conditioning during inference, we employ the classifier-free guidance (CFG) [31] to find $\mathbf{Z}$ that has a high $\log p(\mathbf{c}|\mathbf{Z})$. By Bayes' rule, the gradient of this objective is $\nabla_{\mathbf{Z}} \log p(\mathbf{c}|\mathbf{Z}) \propto \nabla_{\mathbf{Z}} \log p(\mathbf{Z}|\mathbf{c}) - \nabla_{\mathbf{Z}} \log p(\mathbf{Z})$. Following [32], we can interpret the diffusion model $\epsilon_\theta$'s output as score functions, and therefore maximize $p(\mathbf{c}|\mathbf{Z})$ using the modified denoising function:

$$\tilde{\epsilon}_\theta(\mathbf{Z}^{(t)}, t, \mathbf{c}) = (1 + w)\epsilon_\theta(\mathbf{Z}^{(t)}, t, \mathbf{c}) - w\epsilon_\theta(\mathbf{Z}^{(t)}, t), \tag{15}$$

Table 1: Performance of *de novo* 3D molecule generation on GEOM-Drugs. * indicates results reproduced using official source codes, while other baseline results are taken from [6].

| 2D-Metric | FCD↓ | AtomStable | MolStable | V&C | V&U | V&U&N | SNN | Frag | Scaf |
|---|---|---|---|---|---|---|---|---|---|
| Train | 0.251 | 1.000 | 1.000 | 1.000 | 1.000 | 0.000 | 0.585 | 0.999 | 0.584 |
| CDGS | 22.051 | 0.991 | 0.706 | 0.285 | 0.285 | 0.285 | 0.262 | 0.789 | 0.022 |
| JODO | 2.523 | **1.000** | 0.981 | 0.874 | 0.905 | 0.902 | 0.417 | **0.993** | 0.483 |
| MiDi* | 7.054 | 0.968 | 0.822 | 0.633 | 0.654 | 0.652 | 0.392 | 0.951 | 0.196 |
| EQGAT-diff* | 5.898 | **1.000** | **0.989** | 0.845 | 0.863 | 0.859 | 0.377 | 0.983 | 0.161 |
| **UDM-3D, ours** | **0.692** $^{-72.6\%}$ | **1.000** | 0.925 | **0.879** | **0.913** | **0.907** | **0.525** | 0.990 | **0.540** |

| 3D-Metric | FCD$_{3D}$↓ | AtomStable | MolStable | Bond length↓ | Bond angle↓ | Dihedral angle↓ |
|---|---|---|---|---|---|---|
| Train | 13.73 | 0.861 | 0.028 | 1.56E-04 | 1.81E-04 | 1.56E-04 |
| EDM | 31.29 | 0.831 | 0.002 | 4.29E-01 | 4.96E-01 | 1.46E-02 |
| JODO | 19.99 | 0.845 | 0.010 | 8.49E-02 | 1.15E-02 | 6.68E-04 |
| MiDi* | 23.14 | 0.750 | 0.003 | 1.17E-01 | 9.57E-02 | 4.46E-03 |
| GeoLDM | 30.68 | 0.843 | 0.008 | 3.91E-01 | 4.22E-01 | 1.69E-02 |
| EQGAT-diff* | 26.33 | 0.825 | 0.007 | 1.55E-01 | 5.21E-02 | 2.10E-03 |
| **UDM-3D, ours** | **17.36** $^{-13.2\%}$ | **0.852** | **0.014** | **9.89E-03** $^{-88.4\%}$ | **5.11E-03** $^{-55.6\%}$ | **1.74E-04** $^{-74.0\%}$ |

where $w \in [0, +\infty)$ is a hyperparameter controlling the guidance strength; $\epsilon_\theta(\mathbf{Z}^{(t)}, \mathbf{t}, \mathbf{c})$ is the conditional variant of $\epsilon_\theta$, incorporating the property $\mathbf{c}$; $\epsilon_\theta(\mathbf{Z}^{(t)}, \mathbf{t})$ is the unconditioned variant that ignores $\mathbf{c}$. To train $\epsilon_\theta$, we randomly drop condition $\mathbf{c}$ with a certain probability $p_{\text{drop}}$, allowing $\epsilon_\theta$ to learn both conditional and unconditional distributions.

## 4 Experiments

In this section, we present the experimental results of our proposed unified molecular latent space approach (UAE-3D & UDM-3D) on 3D molecule generation tasks. We comprehensively evaluate UDM-3D's performance on *de novo* 3D molecule generation and conditional 3D molecule generation with targeted quantum properties. Our experiments demonstrate significant improvements over state-of-the-art methods while maintaining computational efficiency.

### 4.1 Experimental Setup

**Datasets.** Following prior works [5, 6, 9], we conduct experiments on QM9 [24] and GEOM-Drugs [25] datasets. QM9 [24] contains 130k small organic molecules (up to 9 heavy atoms) with quantum chemical properties. We use 100K/18K/13K train/val/test splits, following [5, 6]. GEOM-Drugs [25] is a pharmaceutical-scale dataset with 450k drug-like molecules (average 44.4 atoms, max 181 atoms). We use 80%/10%/10% splits and retain the lowest-energy conformer per molecule.

**Baselines.** For *de novo* 3D molecule generation, we compare UDM-3D with CDGS [33], JODO [6], MiDi [9], G-SchNet [12], G-SphereNet [34], EDM [5], MDM [35], GeoLDM [17], EQGAT-diff [36], SemlaFlow[37] and ADiT [38]. For conditional generation, we additionally use baselines of EEGSDE [8] and GeoBFN [39]. We do not report MDM's performance on GEOM-DRUGS because we are unable to reproduce reasonable results using their official code. Similarly, ADiT's performance on GEOM-DRUGS is omitted, because it is not available in their paper, code, and released checkpoint.

### 4.2 *De Novo* 3D Molecule Generation

**Experimental Setting.** Generating a 3D molecule involves generating the structural validity (atom/bond features) and accurate spatial arrangements (3D coordinates). Therefore, we adapt the comprehensive metrics from [6, 5]. We also report all the metrics for molecules from training set, serving as an approximate upper bound for performance. Our evaluation metrics can be divided into two groups: (1) **2D Metrics**: Atom stability, validity & completeness (V&C), validity & uniqueness (V&U), validity & uniqueness & novelty (V&U&N), similarity to nearest neighbor (SNN), fragment similarity (Frag), scaffold similarity (Scaf), and Fréchet ChemNet Distance (FCD) [3]; (2) **3D Metrics**: Atom stability, FCD, maximum mean discrepancy (MMD) [40] for the distributions of bond lengths, bond angles, and dihedral angles. More experimental details are in Appendix C.1.

Table 1 and Table 2 present the performance of UDM-3D on the *de novo* generation task on QM9 and GEOM-Drugs datasets, respectively. Our model demonstrates leading performances in generat-

Table 2: Performance of *de novo* 3D molecule generation on QM9. * indicates results reproduced using official source codes, while other baseline results are taken from [6]. Some of ADiTs evaluation metrics are omitted due to differences in evaluation protocols. We discuss these protocol differences and their implications in Appendix C.

| 2D-Metric | FCD↓ | AtomStable | MolStable | V&C | V&U | V&U&N | SNN | Frag | Scaf |
|---|---|---|---|---|---|---|---|---|---|
| Train | 0.063 | 0.999 | 0.988 | 0.989 | 0.989 | 0.000 | 0.490 | 0.992 | 0.946 |
| CDGS | 0.798 | 0.997 | 0.951 | 0.951 | 0.936 | 0.860* | 0.493 | 0.973 | 0.784 |
| JODO | 0.138 | **0.999** | **0.988** | **0.990** | 0.960 | 0.780* | **0.522** | 0.986 | **0.934** |
| MiDi* | 0.187 | 0.998 | 0.976 | 0.980 | 0.954 | 0.769 | 0.501 | 0.979 | 0.882 |
| EQGAT-diff* | 2.088 | **0.999** | 0.971 | 0.965 | 0.950 | 0.891 | 0.482 | 0.950 | 0.703 |
| SemlaFlow | 0.863 | 0.995 | 0.949 | 0.857 | 0.821 | 0.821 | 0.124 | - | - |
| **UDM-3D, ours** | **0.130**$^{-5.80\%}$ | **0.999** | **0.988** | 0.983 | **0.973** | **0.950** | 0.508 | **0.987** | 0.898 |

| 3D-Metric | FCD$_{3D}$↓ | AtomStable | MolStable | Bond length↓ | Bond angle↓ | Dihedral angle↓ |
|---|---|---|---|---|---|---|
| Train | 0.877 | 0.994 | 0.953 | 5.44E-04 | 4.65E-04 | 1.78E-04 |
| EDM | 1.285 | 0.986 | 0.817 | 1.30E-01 | 1.82E-02 | 6.64E-04 |
| MDM | 4.861 | 0.992 | 0.896 | 2.74E-01 | 6.60E-02 | 2.39E-02 |
| JODO | 0.885 | 0.992 | 0.934 | 1.48E-01 | 1.21E-02 | 6.29E-04 |
| GeoLDM | 1.030 | 0.989 | 0.897 | 2.40E-01 | 1.00E-02 | 6.59E-04 |
| MiDi* | 1.100 | 0.983 | 0.842 | 8.96E-01 | 2.08E-02 | 8.14E-04 |
| EQGAT-diff* | 1.520 | 0.988 | 0.888 | 4.21E-01 | 1.89E-02 | 1.24E-03 |
| SemlaFlow | 1.127 | 0.971 | 0.787 | - | - | - |
| ADiT* | 2.884 | 0.211 | - | 9.98E-01 | 3.38E-02 | 1.46E-03 |
| **UDM-3D, ours** | **0.881**$^{-0.45\%}$ | **0.993** | **0.935** | **7.04E-02**$^{-45.8\%}$ | **9.84E-03**$^{-1.6\%}$ | **3.47E-04**$^{-44.8\%}$ |

ing chemically valid and novel molecules and achieves state-of-the-art performance in most of the metrics across both datasets. Two detailed key observations emerge from the results:

**Leading Geometric Accuracy.** Our unified latent space enables highly accurate 3D geometric modeling, a key factor in realistic molecule generation. UDM-3D significantly outperforms baselines in distributional distances of bond lengths and angles, reducing errors by an order of magnitude. These improvements hold even for complex, drug-like molecules in GEOM-Drugswhere UDM-3D achieves a **25x** lower bond length error than GeoLDM (9.89E-03 *v.s.* 3.91E-01). This leap in accuracy comes from UDM-3D's ability to jointly optimize chemical and geometric constraints in a unified latent space, maintaining consistency across molecular modalities.

**Novel and High Quality Generation.** UDM-3D achieves outstanding V&U&N scores of **0.907** on GEOM-Drugs and **0.950** on QM9, while achieving state-of-the-art 3D stability and comparable 2D stability to strong baselines such as JODO. This highlights its ability to generate novel molecules without compromising structural quality. The key lies in the unified latent space, which jointly encodes atom types, bond types, and coordinates, enabling the model to capture complex interdependencies among molecular features.

### 4.3  Conditional 3D Molecule Generation

**Experimental Settings.** We evaluate conditional generation of molecules with target quantum properties using the protocol from [5, 17]. Specifically, our target properties include $C_v$, $\mu$, $\alpha$, $\epsilon_{\text{HOMO}}$, $\epsilon_{\text{LUMO}}$, $\Delta\epsilon$, and we evaluate the Mean Absolute Error (MAE) between target and predicted properties. More details on the properties and settings are provided in Appendix C.2.

Table 3 reports MAE results for conditional generation on QM9. UDM-3D achieves the lowest MAE in five out of six properties, showing strong capability in generating molecules that match target properties. On average, it reduces MAE by 42.2% compared to GeoLDM, with a notable 52.7% improvement in predicting the HOMO-LUMO gap ($\Delta\epsilon$). These results highlight the importance of our unified latent space that effectively integrates all molecular modalities during generation. While GeoLDM struggles to correlate all the modalities due to its fragmented latent spaces, our unified latent representation inherently preserves the interplay between 3D geometry, bonds, and electronic characteristics, enabling precise modeling over molecule-property relationships.

### 4.4  Training & Sampling Efficiency

We further analyze the training and sampling efficiency of UDM-3D compared to baselines on the QM9 dataset. These experiments are performed with an NVIDIA A100 GPU.

Table 3: Mean Absolute Error (MAE) for conditional 3D molecule generation on QM9.

| Method | $\mu$ (D) | $\alpha$ (Bohr$^3$) | $C_v$ $\left(\frac{\text{cal}}{\text{mol}}\text{K}\right)$ | $\varepsilon_{\text{HOMO}}$ (meV) | $\varepsilon_{\text{LUMO}}$ (meV) | $\Delta\varepsilon$ (meV) |
|---|---|---|---|---|---|---|
| U-Bound | 1.613 | 8.98 | 6.879 | 645 | 1457 | 1464 |
| L-Bound | 0.043 | 0.09 | 0.040 | 39 | 36 | 65 |
| EDM | 1.123 | 2.78 | 1.065 | 371 | 601 | 671 |
| EEGSDE | 0.777 | 2.50 | 0.941 | 302 | 447 | 487 |
| GeoLDM | 1.108 | 2.37 | 1.025 | 340 | 522 | 587 |
| GeoBFN | 0.998 | 2.34 | 0.949 | 328 | 516 | 577 |
| JODO | 0.628 | **1.42** | 0.581 | 226 | 256 | 335 |
| **UDM-3D, ours** | **0.603**$^{-3.98\%}$ | 1.54$^{+8.4\%}$ | **0.553**$^{-4.82\%}$ | **216**$^{-4.42\%}$ | **247**$^{-3.5\%}$ | **313**$^{-6.57\%}$ |

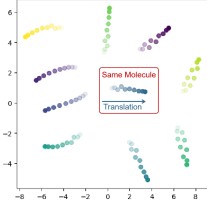

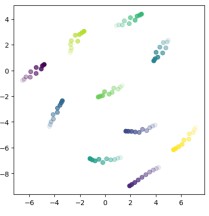

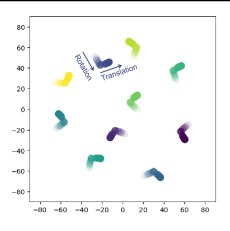

(a) Translation.    (b) Rotation.    (c) Rot. $\rightarrow$ Trans.

Figure 4: t-SNE visualizations of UAE-3D's latents under SE(3) augmentations. (a) Translations along a fixed direction. (b) Rotations along a fixed axis. (c) Sequential rotations followed by translations. Color gradients show increasing distances or angles.

Table 4: Comparing DiT with Transformer and PerceiverIO for latent diffusion modeling. Models have the same depth and hidden size. A.S. $\rightarrow$ atom stability.

| Metric | DiT | Transformer | PerceiverIO |
|---|---|---|---|
| A.S.$_{\text{3D}}$ | **0.993** | 0.983 | 0.972 |
| A.S.$_{\text{2D}}$ | **0.999** | 0.997 | 0.990 |
| V&C | **0.983** | 0.938 | 0.933 |
| V&U | **0.973** | 0.922 | 0.931 |
| V&U&N | **0.950** | 0.922 | 0.931 |

**Training Efficiency.** As Figure 2(c) shows, UDM-3D's training costs 52 hours (14h UAE-3D + 38h UDM-3D), which is 5.3 times faster than GeoLDM and 2.7 times faster than JODO. This efficiency stems from: (1) Our decoupled training paradigm: UAE-3D first learns 3D molecule compression, allowing DiT to focus solely on generative modeling of the compressed latents. (2) Efficient diffusion with DiT: Unlike previous molecular diffusion models that require complex architectures for multi-modality and equivariance, DiT's simple and highly parallelizable design allows for faster training and sampling in the unified latent space. Crucially, our speedup does not compromise performance, as guaranteed by UAE-3D's near-lossless molecular reconstruction (*cf.* Figure 2(a;b)).

**Sampling Speed.** Figure 2(d) shows that UDM-3D generates each molecule in just 0.081s, which is 7.3x faster than EDM/GeoLDM and 9.8x faster than JODO. This speed advantage comes from our unified latent space, which simplifies DiT's modeling and avoids complex neural architectures.

## 4.5    Analysis and Ablation Studies

**UAE-3D's Latents Present Structured Variations across Geometric Movements.** Figure 4 shows the t-SNE visualizations of UAE-3D's latent representation under different SE(3) augmentations. Each dot represents a molecular embedding in the latent space, and the color gradient from light to dark corresponds to an increase in the translation distance (middle) or the rotation angle (right) for the same molecule. We observe that the same molecule's representations remain close in the latent space and as the augmentation scale increases, the representation changes along a consistent direction. Figure 4c shows that the representations change moving directions when the augmentation is switched from rotation to translation. These observations demonstrate that UAE-3D captures meaningful and structured variations in molecular representations across SE(3) augmentations.

**3D Molecule Samples by UDM-3D.** We present 3D molecules generated by our method as case studies in Figure 5. The generated molecules are chemically valid and exhibit diverse and complex structures, showcasing UDM-3D's ability to generate realistic 3D molecular conformations.

**Ablating the DiT Backbone.** We compare DiT with alternative transformer-based architectures for *de novo* 3D molecule generation on the QM9 dataset (*cf.* Table 4). Specifically, we evaluate against a vanilla Transformer and PerceiverIO[41], a modern architecture designed for structured inputs and outputs. The results show that DiT consistently outperforms both baselines across all metrics, which we attribute to its adaptive LayerNorm layers. These layers enable DiT to effectively handle data with varying noise scales, thereby improving diffusion performance. Moreover, when paired with alternative diffusion neural architectures such as Transformer or PerceiverIO, our UAE-3D framework still achieves meaningful and comparable performances, demonstrating its robustness

Table 5: The performance influence of SE(3)-equivariant augmentations on the QM9 dataset.

| 2D-Metric | FCD↓ | AtomStable | MolStable | V&C | V&U | V&U&N | SNN | Frag | Scaf |
|---|---|---|---|---|---|---|---|---|---|
| No aug. | 0.581 | 0.995 | 0.947 | 0.950 | 0.935 | 0.921 | 0.492 | 0.971 | 0.799 |
| + Rot. | 0.315 | 0.995 | 0.948 | 0.951 | 0.943 | 0.927 | 0.493 | 0.979 | 0.875 |
| + Trans. | 0.202 | **0.999** | 0.986 | **0.980** | 0.967 | 0.944 | **0.507** | 0.981 | 0.884 |
| + Trans. + Rot. | **0.130** | **0.999** | **0.988** | **0.983** | **0.973** | **0.950** | **0.508** | **0.987** | **0.898** |

| 3D-Metric | FCD$_{3D}$↓ | AtomStable | MolStable | Bond length↓ | Bond angle↓ | Dihedral angle↓ |
|---|---|---|---|---|---|---|
| No aug. | 1.065 | **0.993** | 0.879 | 1.01E-01 | 1.18E-02 | 1.08E-03 |
| + Rot. | 1.007 | **0.993** | 0.876 | 8.12E-02 | 9.99E-03 | 3.90E-04 |
| + Trans. | 0.902 | 0.989 | 0.896 | 7.08E-02 | 1.04E-02 | 6.81E-04 |
| + Trans. + Rot. | **0.881** | **0.993** | **0.935** | **7.04E-02** | **9.84E-03** | **3.47E-04** |

Table 6: RMSD results when trained with different SE(3) augmentations on QM9.

| Train with | RMSD ($\times 10^{-3}$Å) |
|---|---|
| No aug. | 1.4 |
| + Rot. | 1.1 |
| + Trans. | 0.9 |
| + Trans. + Rot. | **0.2** |

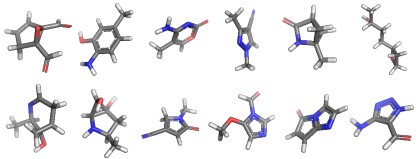

Figure 5: Visualization of random samples generated by UDM-3D on QM9.

to different backbone choices. While further hyperparameter tuning could potentially improve the performance of these alternatives, the results already highlight the advantage of DiT in our setting.

**SE(3)-Equivariant Augmentations.** Table 5 shows the performance influence of various SE(3)-equivariant augmentations on the QM9 dataset. The results reveal that data augmentation improves overall performance by providing diverse geometric views during UDM-3D training.

In addition to the standard performance metrics, we further investigate UAE-3D's reconstruction error on the test set when trained under different 3D transformations. Table 6 presents the RMSD of the reconstructed molecules. By including both rotation and translation augmentations, we observe a significant reduction in reconstruction error (from $1.4 \times 10^{-3}$ to $0.2 \times 10^{-3}$), achieving the lowest RMSD. This shows that our SE(3)-equivariant augmentations effectively enhance the model's ability to learn geometric consistency by exploring a more comprehensive 3D molecular space.

# 5  Related Works

**3D Molecule Generation.** Early efforts for 3D molecule generation focus on autoregressive approaches [12, 13, 34], constructing 3D molecules sequentially by adding atoms or molecular fragments. With the success of diffusion models across various domains [42, 15, 43], they also become the *de facto* method for 3D molecule generation [5, 8]. However, the early diffusion works can easily generate invalid molecules because of overlooking the bond information. To bridge this gap, subsequent works [6, 9, 11] additionally consider bond generation by building a separate diffusion process. However, generating different modalities in separate diffusion processes unnecessarily complicates the model design, reducing efficiency. It also risks compromising the consistency between these modalities. To address this, our UDM-3D performs generative modeling in UAE-3D's unified latent space that integrates all molecular modalities, improving efficiency and generation quality.

**LDMs for 3D Molecule Generation.** For efficiency, LDMs employ a VAE [16] to compress raw data into a low-dimensional latent space, where a diffusion model performs generative modeling [14]. This method has been very popular for image generation [23]. However, existing 3D molecular LDMs still face challenges in separated latent spaces. For example, GeoLDM [17] compresses atom features and 3D coordinates separately, failing to build a unified latent space. Other works [39, 44] process different molecular components (atoms/subgraphs or atom types/coordinates) in separate channels, increasing modeling complexity. In 3D protein generation, PepGLAD [18] compresses 1D sequences and 3D structures into two separate latent spaces. Our UDM-3D addresses these issues by employing a unified latent space, significantly improved generation efficiency.

# 6 Conclusion and Future Works

In this paper, we propose UAE-3D to compress the multi-modal features of 3D molecules into a unified latent space, and demonstrate the effectiveness of latent diffusion modeling on this space by introducing UDM-3D. By integrating atom types, chemical bonds, and 3D coordinates into a single latent space, our model effectively addresses the inherent challenges of multi-modality and SE(3) equivariance for 3D molecule generation. Extensive experiments on GEOM-Drugs and QM9 confirm leading performances in both *de novo* and conditional 3D molecule generation, setting new benchmarks for both quality and efficiency. UAE-3D's success highlights the benefits of building a unified latent space for molecular design. Moving forward, we will transfer this unified latent space to new modalities, including proteins and RNAs, and extend it to broader molecular design tasks, including structure-based drug design. While UAE-3D's latents are near-lossless compressions of the original molecule, it inspires the exploration of jointly modeling UAE-3D's latents with text sequences for text-guided 3D molecule generation, following [45–51], as well as joint modeling of molecular and biological text representations across broader domains, including proteins [52] and single-cell transcriptomics [53].

## Limitations

**Scope.** In this work, we explore 3D molecule generation under the settings of unconditional generation and conditional generation targetting at quantum chemical properties. Other molecule generation tasks, such as structure-based drug design [54], is out of the scope of this work. We leave this exploration to future works.

**Molecule Size.** Our method relies on a separate module to decide each generated molecule's number of atom, following prior works [6, 5, 8]. Specifically, we use the molecule size distribution measured on the training dataset to sample new molecule size. While this method shows decent performance on the tested tasks, we conjecture this will be one of the bottlenecks when the tested task become more complex. Resolving this issue might demand the incorporation of auto-regressive based method for molecule generation, in which the generated molecule size is automatically controlled by the auto-regressive process.

## Acknowledgements

This research/project is supported by the National Research Foundation, Singapore under its National Large Language Models Funding Initiative (AISG Award No: AISG-NMLP-2024-002), the Ministry of Education (MOE T1251RES2309 and MOE T2EP20125-0039), the Agency for Science, Technology and Research (A*STAR) (H25J6a0034), and the National Natural Science Foundation of China (62572449).

Any opinions, findings and conclusions or recommendations expressed in this material are those of the author(s) and do not reflect the views of the National Research Foundation, Singapore.

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

# A   More Experimental Results

## A.1   More Ablation Studies and Analysis

Table 7: The influence of latent space dimensionality for 3D molecule generation on the QM9 dataset.

| Latent Dim | Atom Acc | Bond Acc | RMSD | AtomStable | MolStable | V&C | V&U | V&U&N | Bond Length↓ | Bond Angle↓ | Dihedral Angle↓ |
|---|---|---|---|---|---|---|---|---|---|---|---|
| 4 | 0.9201 | 0.8932 | 0.0080 | 0.914 | 0.325 | 0.580 | 0.566 | 0.566 | 3.27E-01 | 2.20E-01 | 4.19E-03 |
| 8 | 0.9998 | 0.9746 | 0.0006 | 0.987 | 0.882 | 0.942 | 0.923 | 0.923 | 1.29E-01 | 1.77E-02 | 8.14E-04 |
| 16 | 1.0000 | 1.0000 | 0.0002 | 0.999 | 0.988 | 0.983 | 0.973 | 0.950 | 7.04E-02 | 9.84E-03 | 3.47E-04 |
| 32 | 1.0000 | 1.0000 | 0.0003 | 0.986 | 0.867 | 0.943 | 0.918 | 0.918 | 9.98E-02 | 1.42E-02 | 7.77E-04 |

Table 8: The influence of R-Trans layer depth for reconstruction on the QM9 dataset.

| R-Trans layers | Atom Acc | Bond Acc | RMSD |
|---|---|---|---|
| 3 | 1.0000 | 0.9999 | 0.000326 |
| 6 | 1.0000 | 1.0000 | 0.000203 |
| 9 | 1.0000 | 1.0000 | 0.000229 |
| 12 | 1.0000 | 1.0000 | 0.000209 |

Table 9: The influence of different bonded distance loss weight $\lambda$ on the QM9 dataset.

| $\lambda$ / VAE Recon. | Atom Accuracy (%) | | Bond Accuracy (%) | | Coordinate RMSD (Å) | | |
|---|---|---|---|---|---|---|---|
| 0 (w/o $\lambda$) | 1.000 | | 1.000 | | 0.0034 | | |
| 1 | 1.000 | | 1.000 | | 0.0010 | | |
| 5 | 1.000 | | 1.000 | | 0.0007 | | |
| 10 | 1.000 | | 1.000 | | 0.0002 | | |
| 20 | 1.000 | | 1.000 | | 0.0002 | | |

| $\lambda$ / 2D-Metric | FCD↓ | AtomStable | MolStable | V&C | V&U | V&U&N | SNN | Frag | Scaf |
|---|---|---|---|---|---|---|---|---|---|
| 1 | 0.283 | 0.996 | 0.959 | 0.962 | 0.940 | 0.940 | 0.494 | 0.978 | 0.874 |
| 5 | 0.255 | 0.997 | 0.965 | 0.970 | 0.947 | 0.947 | 0.500 | 0.983 | 0.888 |
| 10 | 0.130 | 0.999 | 0.988 | 0.983 | 0.973 | 0.950 | 0.508 | 0.987 | 0.898 |
| 20 | 0.172 | 0.999 | 0.988 | 0.980 | 0.969 | 0.947 | 0.507 | 0.987 | 0.898 |

| $\lambda$ / 3D-Metric | FCD$_{3D}$↓ | AtomStable | MolStable | Bond length↓ | Bond angle↓ | Dihedral angle↓ |
|---|---|---|---|---|---|---|
| 1 | 0.901 | 0.985 | 0.857 | 1.25E-01 | 1.80E-02 | 1.06E-03 |
| 5 | 0.892 | 0.989 | 0.909 | 1.12E-01 | 1.19E-02 | 7.18E-04 |
| 10 | 0.881 | 0.993 | 0.935 | 7.04E-02 | 9.84E-03 | 3.47E-04 |
| 20 | 0.889 | 0.993 | 0.929 | 7.05E-02 | 9.84E-03 | 3.45E-04 |

**Reconstruction Errors.** To assess the reconstruction fidelity of UAE-3D, we evaluate its performance on the held-out test sets of QM9 and GEOM-Drugs. We report atom-type and bond-type accuracies as well as the coordinates RMSD, which jointly reflect the VAE's ability to compress and reconstruct both invariant and equivariant molecular features. Table 11 presents the reconstruction results. On the QM9 dataset, UDM-3D achieves perfect reconstruction accuracy (100% for both atoms and bonds) with a near-zero coordinate RMSD of 0.0002 Å, substantially overperforming Ge-oLDM and ADiT. Notably, ADiT does not retain and reconstruct chemical bond information, which is a limitation when reconstructing 3D molecules. On the more complex GEOM-Drugs dataset, UDM-3D again achieves perfect reconstruction (100.0% atom and bond accuracy, 0.0008 Å RMSD), whereas GeoLDM exhibits noticeable degradation. This demonstrates that UDM-3D's unified latent space not only faithfully compresses all molecular modalities but also generalizes robustly to large and diverse molecular datasets. These results show that UAE-3D effectively learns a unified latent representation that preserves both the chemical identity and geometric precision of molecules. These findings align with the visual comparison in Figure 2(a;b), and reinforce our claim that UAE-3D achieves near-lossless reconstructionan essential prerequisite for accurate latent diffusion modeling.

**Unified vs. Separated Latent Space.** To validate the effectiveness of our unified latent space design, we conduct an ablation study comparing UDM-3D with a variant that employs separate latent spaces for invariant (2D) and equivariant (3D) molecular features. Specifically, we split the unified VAE

Table 10: Comparison between unified and separated latent space on the QM9 dataset.

| 2D-Metric | FCD↓ | AtomStable | MolStable | V&C | V&U | V&U&N | SNN | Frag | Scaf |
|---|---|---|---|---|---|---|---|---|---|
| Unified | 0.130 | 0.999 | 0.988 | 0.983 | 0.973 | 0.950 | 0.508 | 0.987 | 0.898 |
| Separated | 0.351 | 0.995 | 0.952 | 0.943 | 0.920 | 0.913 | 0.341 | 0.940 | 0.682 |

| 3D-Metric | FCD↓ | AtomStable | MolStable | Bond length↓ | Bond angle↓ | Dihedral angle↓ |
|---|---|---|---|---|---|---|
| Unified | 0.881 | 0.993 | 0.935 | 7.04E-02 | 9.84E-03 | 3.47E-04 |
| Separated | 2.356 | 0.982 | 0.872 | 18.4E-02 | 123E-03 | 7.03E-04 |

Table 11: Reconstruction accuracy (%) for atom and bond types, and coordinate RMSD on the test sets.

| Dataset | Method | Atom Accuracy (%) | Bond Accuracy (%) | Coordinate RMSD (Å) |
|---|---|---|---|---|
| GEOM-Drugs | GeoLDM | 96.9 | 93.7 | 0.2526 |
| | UDM-3D, ours | **100.0** | **100.0** | **0.0008** |
| QM9 | GeoLDM | 98.6 | 96.2 | 0.1830 |
| | ADiT | 85.7 | - | 0.3598 |
| | UDM-3D, ours | **100.0** | **100.0** | **0.0002** |

into two independent neural networks: one maintaining the R-Trans architecture and taking 2D information (atom types, bond types) as input, while the other uses a vanilla transformer structure and takes 3D information (atom types and coordinates) as input. Each network generates a separate latent space, with dimensions set to half of the original hyperparameter. The LDM is then trained on the concatenated latent space of these two networks. Table 10 summarizes the results on the QM9 dataset. We observe that UDM-3D with a unified latent space significantly outperforms the separated variant across all 2D and 3D generation metrics. For instance, the FCD score improves from 0.351 to 0.130, and the MolStable metric increases from 0.952 to 0.988. This demonstrates that jointly encoding invariant and equivariant features into a single latent representation enables more effective diffusion modeling, as the LDM can learn a coherent distribution over a unified modality rather than needing to capture complex interactions between two separate modalities. These findings validate our design choice of a unified latent space as crucial for achieving state-of-the-art performance in 3D molecule generation.

**Influenece of Latent Dimensions.** To investigate the impact of latent dimensionality on 3D molecule generation, we conduct an ablation study on the QM9 dataset. Table 7 summarizes the reconstruction errors and the performance metrics across different latent space dimensions (4, 8, 16, and 32). The results indicate that a too-low latent dimension (e.g., 4) fails to capture the necessary information, leading to inferior performance. Increasing the dimension to 16 results in substantial performance improvements, while further increasing to 32 does not yield significant additional gains. This demonstrates that a latent dimension of 16 provides an optimal trade-off between model capacity and reconstruction fidelity.

**Influence of R-Trans Layer Depth.** To assess the effect of depth in the Relational Transformer encoder in UAE-3D, we conduct an ablation study by varying the number of R-Trans layers (3, 6, 9, and 12) on the QM9 dataset regarding reconstruction error. Table 8 summarizes the results. We observe that reducing the number of transformer layers to 3 leads to a noticeable increase in reconstruction error, while increasing it to 9 or 12 does not yield further improvements, as the reconstruction error is already near-zero. This indicates that a moderate depth of 6 layers is sufficient for capturing the necessary relational information among atoms and bonds for effective reconstruction.

**Bonded Distance Loss.** Table 9 presents an ablation study on the weight $\lambda$ for the bonded distance loss on the QM9 dataset. We vary $\lambda$ to analyze its effect on reconstruction accuracy and the preservation of bond geometries. The results indicate that setting $\lambda = 10$ achieves an optimal balance between compressing coordinate information and maintaining geometric fidelity. This experiment validates the necessity of tuning the bonded distance loss weight to enhance model performance.

**Ring Distribution.** Following the evaluation protocol of [54], we computed the Percentage (%) of molecular modes in terms of ring distribution for our generated molecules and compared them

Table 12: Comparison of ring distributions between molecules in the QM9 training set and those generated by our model.

|  | 3-Ring | 4-Ring | 5-Ring | 6-Ring | (7+)-Rings |
|---|---|---|---|---|---|
| QM9 train set | 43.14% | 39.20% | 39.31% | 12.72% | 2.92% |
| Ours | 41.69% | 38.20% | 36.37% | 12.90% | 3.13% |

against the QM9 training set. The results in Table 12 provide a quantitative assessment of how well our model preserves the structural diversity of the training dataset.

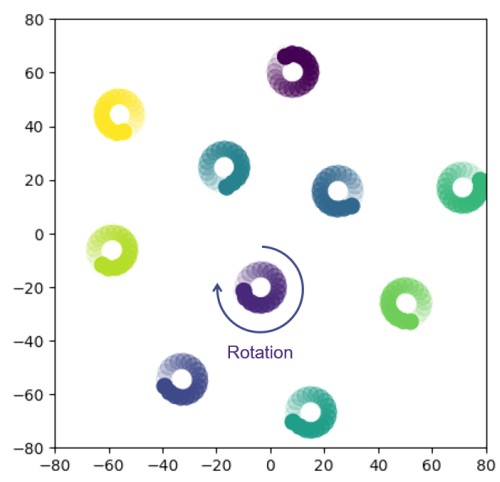 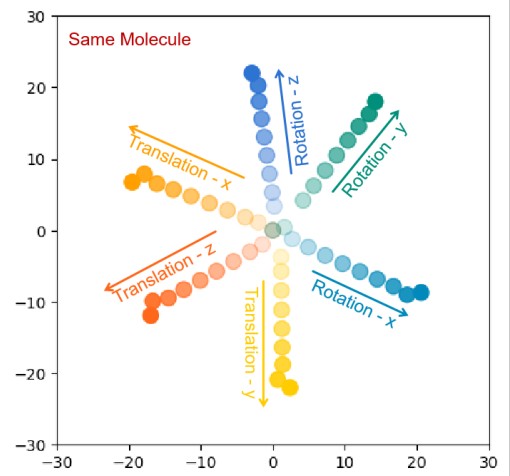

Figure 6: Additional t-SNE visualizations of UAE-3D's latents under SE(3) augmentations. (a) 360-degree rotation along the z-axis. (b) same molecule under different SE(3) augmentations.

**More Visualizations of Latent Space.** In addition to Figure 4, we conducted more t-SNE visualizations of UAE-3D's molecular latent representations to further examine its geometric sensitivity and consistency. In Figure 6(a), we apply 20-step rotations along the same axis (z-axis) to different molecules, completing a full 360-degree rotation. Remarkably, the latent trajectories of all molecule form loops and return to their original positions, reflecting the geometric sensitivity rotational consistency in the learned representations. In Figure 6(b), each color now corresponds to a different SE(3) augmentation applied to the same molecule, rather than to different molecules as in previous plots. We observe clear separation among representations generated by translations and rotations along different axes (x/y/z), indicating that UAE-3D effectively distinguishes between types of geometric transformations in its latent space. These results further validate the structured and disentangled nature of UAE-3D's latent geometry.

### A.2 More Visualizations of Generated Molecules

To further demonstrate the chemical validity and structural diversity of molecules generated by UDM-3D, we provide extended visual comparisons across the GEOM-Drugs and QM9 datasets in Figure 7 and Figure 8.

## B More Related Works

**Molecular Variational Autoencoders.** Variational Autoencoders (VAEs) have been widely explored for generative modeling [16, 55–57], providing a framework to encode data into a structured latent space while enabling both generation and reconstruction [16]. In molecular generation, VAEs have been applied to 2D molecular graphs, as seen in JT-VAE [1], MGCVAE [58], SSVAE [59], and CGVAE [60]. However, these methods have several limitations. First, they primarily focus on 2D

molecular representations and often rely on non-transformer architectures, limiting their scalability for more complex molecular generation tasks [61]. Additionally, VAE-based methods typically assume a simple Gaussian prior, which may fail to accurately model the complex posterior distributions required for effective molecular generation [62, 63]. Notably, masked graph modeling [64] also builds molecular a graph auto-encoder, but aiming for representation learning, instead of molecule generation. With recent advancements in Latent Diffusion Models (LDMs) demonstrating success across multiple domains [65, 66], LDMs have been introduced for 3D molecule and protein generation, as seen in GeoLDM[17] and PepGLAD[18]. These approaches leverage diffusion modeling in latent space, offering improved generative capacity and efficiency.

**Comparison to Other Unified Generative Molecule Models.** Concurrent with our work, ADiT [38] from FAIR also explores latent diffusion for 3D molecules using a variational autoencoder to compress structures into a unified latent space. However, our approach differs in three key ways: (1) **ADiT does not achieve near-lossless compression.** In contrast, we explicitly evaluate reconstruction quality and show significantly lower errors than ADiT (*cf.* Table 11), which translates to better 3D generation performance (*cf.* Table 2). (2) **ADiT omits chemical bonds as part of the inputs and reconstruction targets, leading to information loss.** Although bond types can be inferred from 3D coordinates, such predictions are not 100% accurate. For example, the QM9 dataset [24] reports 3% inconsistency across inference methods. To address this, we explicitly model bonds using a relational transformer [21] and include bond reconstruction as an objective, offering a more robust and chemically faithful modeling approach. (3) **ADiT provides insufficient evaluation for small molecule generation.** ADiT reports only two metrics (validity and uniqueness) for the QM9 dataset, while our evaluation includes 15 metrics, covering 2D structure, 3D geometry, stability, validity, uniqueness, and other critical aspects (*cf.* Table 1). We conjecture that this is because ADiT is more focused on crystal material generation, instead of small molecule generation.

MolCRAFT [54] also aims to unify multiple molecular modalities for generation. However, it processes atom types and coordinates separatelyconcatenating them while use special design to preserve SE(3) equivariance for coordinates. In contrast, our model encodes atom types, coordinates, and bond types into a single unified latent representation, enabling effortless multi-modal modeling using a uni-modal latent diffusion model – Diffusion Transformer [23]. Moreover, MolCRAFT focuses on the task of structure-based drug design, which is different than our major benchmarks.

**Comparison to Other Equivariant 3D Molecule Generation Methods.** We adopt SE(3) data augmentations to train a variational autoencoder without explicitly incorporating equivariant architectures, allowing the model to learn geometric symmetries directly from data. This strategy is motivated by AlphaFold3 [28], which demonstrated that SE(3) augmentations are sufficient for general-purpose networks to learn equivariance and invariance. As shown in Table 6, our model achieves near-zero reconstruction errors on 3D molecules subjected to random rotations and translations, showing successful learning of SE(3) equivariance. Compared to models with built-in SE(3) equivariance, such as EDM [19] and JODO [6], our method avoids the architectural complexities of handling multi-modality and enforcing SE(3) equivariant constraints. This design choice allows us to use the highly optimized and parallelizable diffusion model, Diffusion Transformer [23], for latent diffusion modeling. Consequently, we achieve significant efficiency gains, with up to 8.6x faster training and 9.8x faster inference (*cf.* Figure 2(c;d)).

# C  More Experimental Settings

## C.1  Experimental Settings for *De Novo* 3D Molecule Generation

**Setup.** For the GEOM-Drugs dataset, we use only the ground-state conformer with the lowest energy for each molecule, following the setup in JODO [6]. We adopt the same train/validation/test split ratio of 8:1:1, ensuring a fair comparison. For the QM9 dataset, we use the 100K/10K/10K train/validation/test split. For both datasets, all atoms are shifted to zero center-of-mass before input. All baseline results are either directly borrowed from JODO [6] or reproduced using their released codebase under identical settings.

**Hyperparameters.** 13 shows the key hyperparameters used for training the UAE-3D and UDM-3D models.

Table 13: Hyperparameters of the UAE-3D and UDM-3D models.

| Parameter | UAE | UDM |
|---|---|---|
| epochs | 2000 | 10000 |
| atom loss weight ($\gamma_{\text{atom}}$) | 1.0 | - |
| bond loss weight ($\gamma_{\text{bond}}$) | 1.0 | - |
| coordinate loss weight ($\gamma_{\text{coordinate}}$) | 1.0 | - |
| distance loss weight ($\gamma_{\text{distance}}$) | 1.0 | - |
| bonded distance loss weight ($\lambda$) | 10.0 | - |
| KLD loss weight | 1e-8 | - |
| batch size | 512 | |
| optimizer | AdamW | |
| learning rate | 1e-4 | |
| weight decay | 1e-5 | |
| translation augmentation scale | 0.1 | |
| encoder hidden size | 64 | |
| encoder #heads | 8 | |
| encoder #blocks | 6 | |
| latent dimension | 16 | |
| decoder hidden size | 64 | |
| decoder #heads | 8 | |
| decoder #blocks | 4 | |
| diffusion hidden size | - | 512 |
| diffusion #heads | - | 8 |
| diffusion #layers | - | 8 |
| diffusion mlp ratio | - | 4.0 |
| condition drop ($p_{\text{drop}}$) | - | 0.1 |

**Evaluation Metrics.** Our evaluation framework for *de novo* 3D molecule generation encompasses complementary metrics to assess molecular validity, diversity, and geometric fidelity. These metrics operate at two levels:

**2D Structural Analysis:**

- *Validity & Stability*: Measures adherence to chemical rules:
  - Atom Stability: Percentage of atoms with chemically valid valency, determined by cross-referencing the bond counts with allowed configurations for each atom type (*e.g.,* carbon with 4 bonds). This is evaluated using RDKit's atom valency parser. Formal charges are taken into account in the valency specification.
  - Molecule Stability: Percentage of molecules with valid valency for all atoms pass the above valency check.
  - Validity & Completeness (V&C): Fraction of fully connected, syntactically correct (SMILES-parsable) molecules, excluding fragmented or hypervalent structures.
- *Diversity*: Quantifies exploration of chemical space:
  - Validity & Uniqueness (V&U): Percentage of unique molecules among valid ones, calculated as the ratio of unique SMILES strings to the total number of valid molecules.
  - Validity & Uniqueness & Novelty (V&U&N): Percentage of unique molecules among valid ones that are also novel, calculated as the ratio of unique SMILES strings to the total number of valid molecules, excluding those present in the training set.
- *Distribution Alignment*: Convert all the valid molecules into SMILES strings and compare generated/test distributions on the MOSES benchmark [3]:
  - Fréchet ChemNet Distance (FCD): Similarity between generated and reference distributions using activations from the penultimate layer of ChemNet [67], a pretrained molecular property predictor. Lower scores indicate better alignment.
  - Similarity to the nearest neighbor (SNN): Tanimoto similarity between the generated molecules and their nearest neighbors in the test set, calculated using Morgan fingerprints.
  - Fragment Similarity (Frag): Tanimoto similarity between the generated molecules and their nearest neighbors in the test set, calculated using BRICS [68] fragments.

– Scaffold Similarity (Scaf): Frequencies of Bemis-Murcko scaffolds [69] in the generated molecules compared to the test set.

**3D Geometric Analysis:**

- *Conformer Quality*: The 3D coordinates of generated molecules are converted into bond graphs using a predefined lookup of atomic distance thresholds (per atom pair). Based on these bonds, 2D molecular graphs are reconstructed and then evaluated for atom and molecule stability using the same criteria as in 2D Structural Analysis.
    - Atom Stability: Consistency of valency in 3D-derived 2D graphs. Reconstructs 2D bond orders from 3D atomic distances (via tabulated bond-length thresholds) and checks valency compliance.
    - Molecule Stability: Percentage of molecules with valid valency for all atoms pass the above valency check.
    - $FCD_{3D}$: Same computation as standard FCD, but applied to the reconstructed molecules from 3D coordinates. Captures distributional similarity under geometric constraints.
- *Geometric Fidelity*: Measures spatial arrangement accuracy via Maximum Mean Discrepancy (MMD) for:
    - Bond lengths: MMD between distributions of bond lengths for eight frequent bond types (*e.g.,* C-C, C-N, C-O). Computed using Gaussian kernel with bandwidth tuned via the median heuristic.
    - Bond angles: MMD between the bond angle distributions of generated vs. reference molecules. Triplets of bonded atoms (*e.g.,* C-C-C) define each angle.
    - Dihedral angles: MMD between distributions of dihedral (torsional) angles over four-atom sequences (*e.g.,* C-C-C-C). Captures conformational realism.

**Comparison to Other Evaluation Protocols.** Discrepancies in evaluation methodologies can significantly impact reported performance metrics. For instance, ADiT's validity scores are obtained by reconstructing molecules from atomic coordinates using `pymatgen.core.Molecule`, followed by bond inference via RDKit (likely using the `xyz2mol` heuristic [70]). This pipeline ensures topological consistency (i.e., SMILES validity) but does not explicitly enforce chemical stability constraints such as correct valency or bond geometry. In contrast, our evaluation employs RDKit's `Chem.Mol` for molecule reconstruction, which enforces stricter valency checks and chemical correctness. And our broad evaluation criteria comprehensively measure the validity, diversity, and geometric fidelity of 3D molecule generation.

## C.2 Experimental Setting for Conditional 3D Molecule Generation

**Setup.** We conduct conditional molecule generation on the QM9 dataset following the protocol in EDM [5] and EEGSDE [8]. The properties are evaluated by the QM9 property predictor [24] pre-trained on half of the QM9 dataset. We use the property classifier pre-trained by EDM [5].

Specially, we optimize molecules toward six key electronic properties from quantum chemistry for conditional generation:

- *Electronic Response*:
    - Dipole Moment ($\mu$): Molecular polarity measure
    - Polarizability ($\alpha$): Induced dipole response to electric fields
- *Thermodynamic Properties*:
    - Heat Capacity ($C_v$): Energy absorption at constant volume
- *Electronic Structure*:
    - HOMO ($\varepsilon_{\text{HOMO}}$)/LUMO ($\varepsilon_{\text{LUMO}}$): Frontier orbital energies
    - HOMO-LUMO Gap ($\Delta\varepsilon$): Critical for reactivity and conductivity

The evaluation protocol follows a rigorous split-and-validate strategy [5]:

- QM9 training data divided into disjoint subsets $D_a$ (50k) and $D_b$ (50k)
- Property predictor $\phi_c$ trained exclusively on $D_a$

- Generated molecules compared against $\phi_c$'s predictions on $D_b$ (lower-bound baseline) and random predictions without any relation between molecule and property (upper-bound baseline)
- Performance quantified via Mean Absolute Error (MAE) across all properties

This approach ensures fair assessment of conditional generation without data leakage, with $\phi_c$'s $D_b$ performance establishing the theoretical minimum achievable error.

---

**Algorithm 1** Training Algorithm for UAE-3D and UDM-3D

---

**Require:** Molecular dataset $\mathcal{D}$, encoder $\mathcal{E}$, decoder $\mathcal{D}$, diffusion model $\epsilon_\theta$
1: Initialize encoder $\mathcal{E}$, decoder $\mathcal{D}$, and diffusion model $\epsilon_\theta$
2: **Stage 1: Train UAE-3D**
3: **while** not converged **do**
4:     Sample a batch of 3D molecules $\mathbf{G} \sim \mathcal{D}$
5:     $\mathbf{Z} = \mathcal{E}(\mathbf{G})$ {Encode}
6:     $\hat{\mathbf{G}} = \mathcal{D}(\mathbf{Z})$ {Decode}
7:     $\mathcal{L}_{\text{recon}}$ (Eq. 12) {Reconstruction loss}
8:     $\mathcal{L}_{\text{UAE-3D}}$ (Eq. 13) {VAE loss}
9:     Update $\mathcal{E}$ and $\mathcal{D}$ using $\mathcal{L}_{\text{UAE-3D}}$
10: **end while**
11: **Stage 2: Train UDM-3D**
12: **while** not converged **do**
13:     Sample a batch of latent sequences $\mathbf{Z}^{(0)} \sim \mathcal{E}(\mathcal{D})$
14:     Sample diffusion timestep $t \sim \text{Uniform}(0, 1)$
15:     $\mathbf{Z}^{(t)} = \sqrt{\bar{\alpha}^{(t)}}\mathbf{Z}^{(0)} + \sqrt{1 - \bar{\alpha}^{(t)}}\epsilon$, where $\epsilon \sim \mathcal{N}(\mathbf{0}, \mathbf{I})$ {Diffusion}
16:     $\hat{\epsilon} = \epsilon_\theta(\mathbf{Z}^{(t)}, t)$
17:     $\mathcal{L}_{\text{diffusion}} = \|\hat{\epsilon} - \epsilon\|^2$ {Noise prediction loss}
18:     Update $\epsilon_\theta$ using $\mathcal{L}_{\text{diffusion}}$
19: **end while**

---

**Algorithm 2** UDM-3D Sampling with CFG

---

**Require:** Trained UAE-3D encoder $\mathcal{E}$, DiT $\epsilon_\theta$
**Require:** Guidance strength $w$, timesteps $\{t_k\}_{k=1}^K$
**Require:** Target property $c$ (optional)
1: $\mathbf{Z}^{(1)} \sim \mathcal{N}(0, \mathbf{I})$ {Initial noise}
2: **for** $k = K$ **downto** 1 **do**
3:     $t \leftarrow t_k$
4:     **Compute Guidance:**
5:     $\epsilon_{\text{cond}} \leftarrow \epsilon_\theta(\mathbf{Z}^{(t)}, t, c)$
6:     $\epsilon_{\text{uncond}} \leftarrow \epsilon_\theta(\mathbf{Z}^{(t)}, t, \emptyset)$
7:     $\tilde{\epsilon} \leftarrow (1 + w)\epsilon_{\text{cond}} - w\epsilon_{\text{uncond}}$ {CFG blending}
8:     **Denoise:**
9:     $\mathbf{Z}^{(t-1)} \leftarrow \text{DDPM\_Step}(\mathbf{Z}^{(t)}, \tilde{\epsilon}, t)$ [15]
10: **end for**
11: **Decode:**
12: $\hat{\mathbf{G}} = \langle \hat{\mathbf{F}}, \hat{\mathbf{E}}, \hat{\mathbf{X}} \rangle \leftarrow \mathcal{D}(\mathbf{Z}^{(0)})$ {UAE-3D decoder}

---

### C.3 Pseudo Code

We provide the training and sampling algorithms for UDM-3D in Algorithms 1 and 2, respectively. The training algorithm for UDM-3D involves two stages: training the VAE (UAE-3D) and training the DiT (UDM-3D). The sampling algorithm for UDM-3D with CFG involves iteratively denoising the latent sequence $\mathbf{Z}$ using the DiT and the CFG guidance.

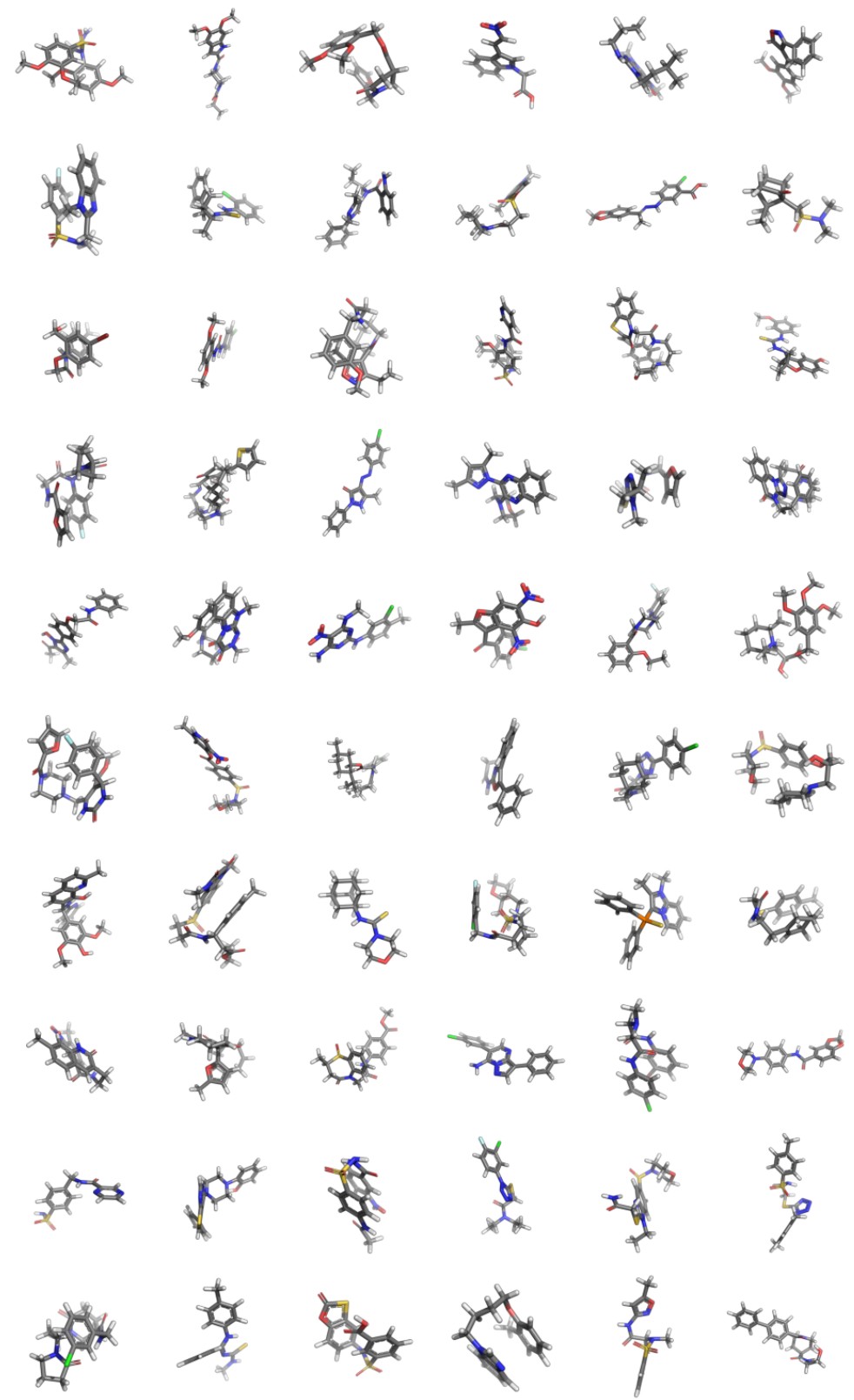

Figure 7: Visualization of random samples generated by UDM-3D on GEOM-Drugs.

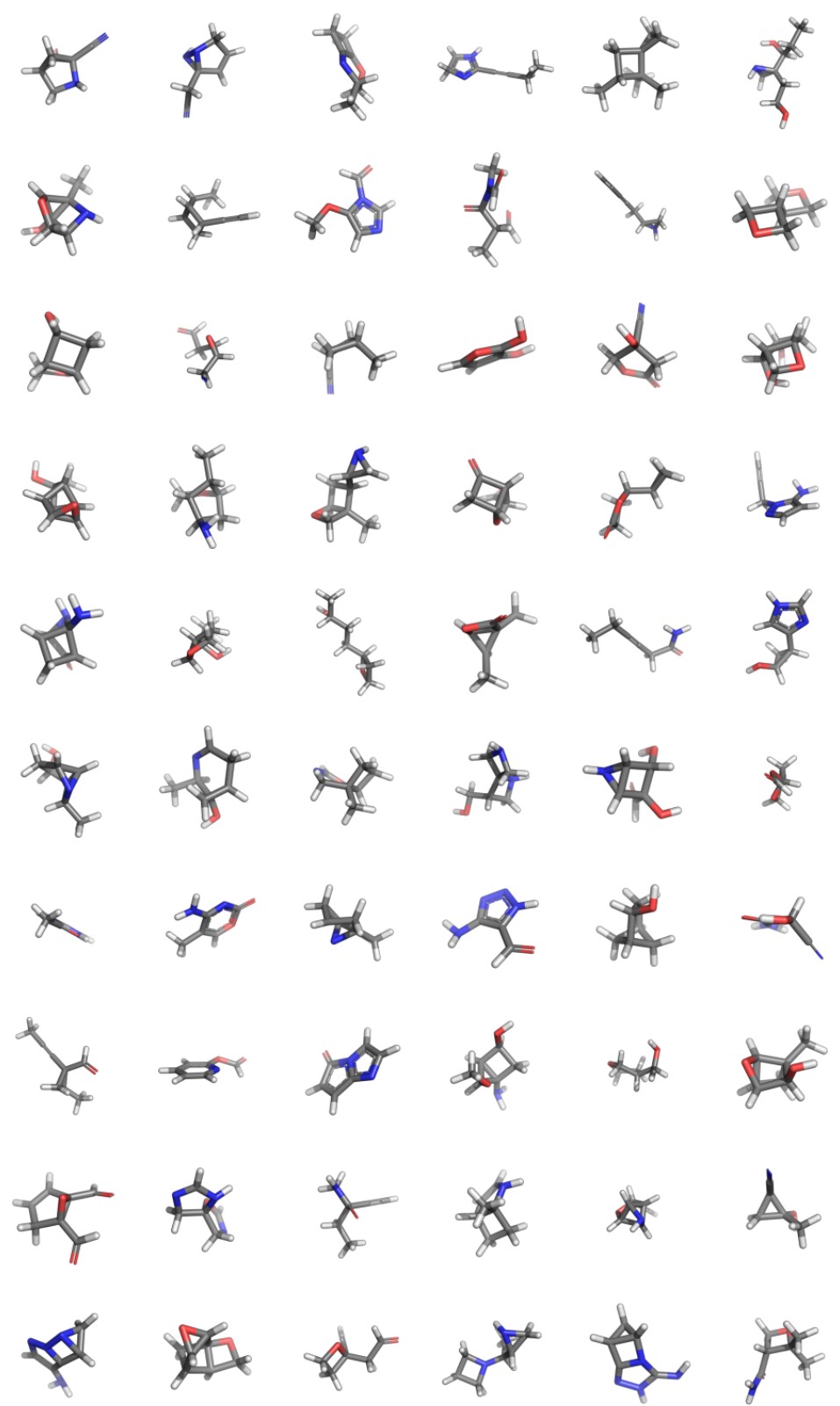

Figure 8: More visualization of random samples generated by UDM-3D on QM9.

