# OpenReview forum: "Towards Unified and Lossless Latent Space for 3D Molecular Latent Diffusion Modeling"
_NeurIPS.cc/2025/Conference — NeurIPS 2025 poster_

### Official Review · Reviewer_eGEZ · 2025-06-21

**Clarity:** 3
**Significance:** 2
**Originality:** 3
**Rating:** 5
**Confidence:** 2

**Summary:**

Molecules possess multiple modalities, such as atom types, inter-atomic bonds, and 3D spatial positions. Latent-based generative molecular models typically use distinct latent spaces to capture different properties like equivariance and invariance. In this manuscript, the authors propose a generative model that utilizes a single latent space—embedding all modalities within the same representation used for a diffusion process. The proposed method is faster than existing approaches while maintaining great accuracy.

**Questions:**

- In the section SE(3)-Equivariant Augmentations, the authors could cite other works that use similar data augmentation techniques—for instance, AlphaFold 3.

- The manuscript refers to both Lossless and near-Lossless reconstruction. I slightly prefer the term near-Lossless, as any trained method may result in minor reconstruction losses.

- For the sampling time comparisons: did you re-run each model (e.g., GeoLDM), or are the reported sampling times taken from their original papers?

- I would be interested in further exploration of the learned latent space. For example, does it preserve molecular clusters?

- What variant of Maximum Mean Discrepancy (MMD) do you use? The result depends on the kernel choice.

*Minor Comments:*

> Equivariance through our tailor-made SE(3) augmentations

Unless the authors introduce novel augmentations, this phrasing may be overstated, as these data augmentation strategies are fairly common in small molecule and protein design.

> ** indicates results reproduced

This sentence should not begin with a symbol.

**Ethical Concerns:**

["NO or VERY MINOR ethics concerns only"]

**Final Justification:**

I would like to thank the authors for engaging in the discussion. I think the new experiments improve the quality of the manuscript.

**Limitations:**

The limitations of the work are not highlighted—either through experiments or in an explicit discussion. The authors should clarify and discuss limitations more clearly.

**Paper Formatting Concerns:**

I did not notice any formatting issues.

**Quality:**

2

**Strengths And Weaknesses:**

**Strengths**

The authors provide extensive comparisons against many baseline methods. The proposed model appears faster while achieving comparable or superior results to existing approaches.

**Weaknesses**

On GEOM-Drugs results:

- JODO should be bold for the Frag metric.

On QM9 results:

- JODO should be bold in Table 2 for the SNN metric.
- In Table 2, EQGAT-diff should be bold for the AtomStable metric (2D).

Overall, most results in Table 2 are very close, especially for the 2D metrics.

It is difficult to assess whether the model truly performs better for de novo design, as many scores are quite similar to those of JODO. If the authors are aware of another dataset that better differentiates between methods, additional experiments would be appreciated. On the other hand, since most studies use these datasets, the benchmark might be saturated. Still, the results are interesting, as they suggest that a different modeling paradigm can achieve comparable or better outcomes. I would suggest revising phrasing such as "Breakthrough in Geometric Accuracy," as this claim is not clearly supported by the results in Tables 1 and 2. It is perfectly acceptable that the method does not outperform across all metrics—what is compelling is that a different architecture and training paradigm can achieve competitive results while being more efficient. This, in itself, can be a valuable contribution.

---

> ### Author Rebuttal · Authors · 2025-07-31
>
> > (W1) On GEOM-Drugs results: JODO should be bold for the Frag metric.
>
> Thank you for the gentle reminder. We have fixed these in the revision.
>
> > (W2) On QM9 results: JODO should be bold in Table 2 for the SNN metric. In Table 2, EQGAT-diff should be bold for the AtomStable metric (2D). Overall, most results in Table 2 are very close, especially for the 2D metrics.
>
> Thank you for pointing this out. We have corrected the boldface formatting of Table 2 accordingly: JODO is now bolded for the SNN metric, and EQGAT-diff is bolded for AtomStable (2D). We also agree that most 2D metrics are close, which further emphasizes the necessity to explore beyond 2D geometry and evaluate holistic 3D generative fidelity.
> We also want to emphasize that our method shows significantly better performance on 3D metrics, especially Bond Length, Bond Angle, and Dihedral Angle on the both datasets, indicating that our method achieves higher accuracy and reliability in 3D molecular generation.
>
> > (W3) It is difficult to assess whether the model truly performs better for de novo design, as many scores are quite similar to those of JODO. If the authors are aware of another dataset that better differentiates between methods, additional experiments would be appreciated. On the other hand, since most studies use these datasets, the benchmark might be saturated. Still, the results are interesting, as they suggest that a different modeling paradigm can achieve comparable or better outcomes. I would suggest revising phrasing such as "Breakthrough in Geometric Accuracy," as this claim is not clearly supported by the results in Tables 1 and 2. It is perfectly acceptable that the method does not outperform across all metrics—what is compelling is that a different architecture and training paradigm can achieve competitive results while being more efficient. This, in itself, can be a valuable contribution.
>
> Thank you for the appreciation. We agree and have toned down the statement of "Breakthrough in Geometric Accuracy" to "Leading Geometric Accuracy" to better reflect the empirical findings. While our method does not always dominate the benchmarks, it demonstrates improved or competitive  performance under a novel modeling paradigm. We believe the introduction of a unified, lossless latent space—combined with efficient training—constitutes a meaningful contribution to the field.
>
> We also acknowledge the potential saturation of current benchmarks and are actively exploring more challenging and discriminative datasets for future work.
>
> > (Limitations) The limitations of the work are not highlighted—either through experiments or in an explicit discussion. The authors should clarify and discuss limitations more clearly.
>
> The limitation section is now in Appendix A due to our limited space in main paper. We appreciate the advice, and will revise to include it in the main paper once have more space in the camera-ready version.
>
> > (Q1) In the section SE(3)-Equivariant Augmentations, the authors could cite other works that use similar data augmentation techniques—for instance, AlphaFold 3.
>
> Thank you for the suggestion. We have included references to prior works such as AlphaFold 3 [2] and MCF [1] that also employ SE(3)-equivariant augmentations in Section 3.1 of the revised manuscript.
>
> > (Q2) The manuscript refers to both Lossless and near-Lossless reconstruction. I slightly prefer the term near-Lossless, as any trained method may result in minor reconstruction losses.
>
> We appreciate this semantic clarification. In the revised version, we will consistently use the term *near-lossless* to acknowledge the minimal but nonzero reconstruction loss.
>
> > (Q3) For the sampling time comparisons: did you re-run each model (e.g., GeoLDM), or are the reported sampling times taken from their original papers?
>
> All reported sampling times are obtained by re-running the official implementations of baselines (e.g., GeoLDM, EDM) under the same hardware and sampling settings to ensure fair comparison.
>
> > (Q4) I would be interested in further exploration of the learned latent space. For example, does it preserve molecular clusters?
>
> We appreciate your insightful suggestion. We have clustered the molecules in the test set based on their Scaffold and embedded all molecules into the unified latent space, resulting in a t-sne visualization. We observe that molecules with the same scaffolds are relatively close in the latent space, showing that our unified latent space effectively captures molecular structure. Unfortunately, we are unable to provide this visualization in image form at this rebuttal stage, but we will include it in the final version.
>
> > (Q5) What variant of Maximum Mean Discrepancy (MMD) do you use? The result depends on the kernel choice.
>
> We follow the same kernel and implementation as in baseline works (e.g., EDM, JODO), using an RBF kernel with a fixed bandwidth. For completeness, we will provide these details in the revised Appendix.
>
> > Equivariance through our tailor-made SE(3) augmentations: unless the authors introduce novel augmentations, this phrasing may be overstated, as these data augmentation strategies are fairly common in small molecule and protein design.
>
> We appreciate this note. While our SE(3)-equivariant augmentations follow standard practice, we emphasize that their integration into both encoding and diffusion stages contributes to the equivariance of the unified latent space. We have revised the phrasing to avoid overstatement and better reflect the novelty in usage rather than in augmentation design per se.
>
> > \* indicates results reproduced – this sentence should not begin with a symbol.
>
> Thank you for pointing this out. We will rewrite these sentences as *The results marked with an asterisk (\*) are reproduced using official source codes*.
>
> Reference:
>
> [1] Swallowing the Bitter Pill: Simplified Scalable Conformer Generation, Wang et al., ICML 2024.
>
> [2] Accurate structure prediction of biomolecular interactions with AlphaFold 3, Abramson et al., Nature 2024.

---

> > ### Comment · Reviewer_eGEZ · 2025-08-04
> >
> > Thank you for addressing my comments and answering my questions. I don't have additional questions at the moment, but I will keep following the discussion with other reviewers.

---

> > > ### Author Response · Authors · 2025-08-05
> > > **Thanks for the feedback**
> > >
> > > Thank you very much for acknowledging our rebuttal and for your thoughtful feedback. We appreciate your positive feedbacks
> > > and we are glad that our responses have addressed your comments and questions. Your constructive suggestions—particularly on phrasing, benchmark interpretation, and additional analysis—have been very helpful and insightful in refining our work. We will incorporate these refinements in the final version.

---

### Official Review · Reviewer_78B5 · 2025-07-02

**Clarity:** 3
**Significance:** 2
**Originality:** 3
**Rating:** 5
**Confidence:** 4

**Summary:**

The paper describes a two step approach to molecular generation where first a VAE (built using transformers) is trained to learn a unified per-atom latent space of 3D molecules and then a diffusion model is trained to generate such spaces. The model demonstrates success in both unconditional and conditional evaluations and is significantly more efficient than the compared approaches.

**Questions:**

How do you achieve near zero reconstruction with a VAE? The latent space is variational so one would expect sampling from the generated embedding to produce different outputs. Do you only evaluate the mean? What are sampled outputs like?

To what extent is this model permutation invariant?

Can you design an ablation study or other experiment that supports your claim the accuracy of your model is due to the unified latent space instead of any of the other architectural and training procedure choices you made?

What advantages does your approach have over recently described flow matching approaches?

**Ethical Concerns:**

["NO or VERY MINOR ethics concerns only"]

**Final Justification:**

The additional evaluations greatly strengthen the paper - they show they outperform a more relevant SotA baseline and perform an ablation study demonstrating the effect of their choice of latent space.

**Limitations:**

yes

**Quality:**

3

**Strengths And Weaknesses:**

Strengths.  The described approach is highly efficient as diffusion can be done in the unified latent space using standard architectures.  The paper is clearly written and the model makes interesting use of transformers.

Weaknesses.
This is a fast moving field and the SoA has advanced beyond the presented baselines (e.g., SemlaFlow outperforms the presented work both in terms of accuracy and speed).

The performance of the model is repeatedly attributed to the unified latent space, but no evidence is provided supporting this assertion.

---

> ### Author Rebuttal · Authors · 2025-07-31
>
> > (W1) This is a fast moving field and the SoA has advanced beyond the presented baselines (e.g., SemlaFlow outperforms the presented work both in terms of accuracy and speed).
> >
> > (Q4) What advantages does your approach have over recently described flow matching approaches?
>
> We thank the reviewer for pointing out recent advances. In response, we have added a comparison to SemlaFlow, a recent baseline. As shown in the following Table, our model outperforms SemlaFlow in both sampling speed and generation quality. The results are reproduced using the official code and pre-trained model checkpoints and evaluated with the same evaluation metrics as our model.
> Note that the missing metrics in SemlaFlow are due to the frequent NaN results in the calculation of related metrics caused by the existing errors in the generated molecules, making it difficult to compute reasonable reporting numbers.
> We will further organize and analyze the results of SemlaFlow, and include this comparison and analysis in the revised version.
>
> Table: 2D Evaluation metrics on the QM9 dataset.
> | Method | FCD$\downarrow$ | AtomStable | MolStable | V&C | V&U | V&U&N | SNN | Frag | Scaf |
> |---|---|---|---|---|---|---|---|---|---|
> | SemlaFlow | 0.863 | 0.995 | 0.949 | 0.857 | 0.821 | 0.821 | 0.124 | - | - |
> | Ours | 0.130 | 0.999 | 0.988 | 0.983 | 0.973 | 0.950 | 0.508 | 0.987 | 0.898 |
>
> Table: 3D evaluation metrics on the QM9 dataset.
> | Method | FCD$\downarrow$ | AtomStable | MolStable | Bond length$\downarrow$ | Bond angle$\downarrow$ | Dihedral angle$\downarrow$ |
> |---|---|---|---|---|---|---|
> | SemlaFlow | 1.127 | 0.971 | 0.787 | - | - | - |
> | Ours | 0.881 | 0.993 | 0.935 | 7.04E-02 | 9.84E-03 | 3.47E-04 |
>
> Table: Sampling speed comparison on the QM9 dataset.
> | Method | Sampling Time (s/molecule) |
> |---|---|
> | SemlaFlow | 0.993 |
> | Ours | 0.081 |
>
> Our approach offers two key advantages over recent flow-matching models such as SemlaFlow:
>
> - **Better Generation Quality**: Our model achieves significantly better generation quality across multiple metrics, including FCD, AtomStable, MolStable, and others. This is due to our unified latent space design that captures both 2D and 3D molecular features effectively.
> - **Improved sampling speed**: Our model achieves **10x faster sampling** compared to SemlaFlow, as shown in the above table. This is due to our latent diffusion model (LDM) design, which allows for efficient sampling in a unified latent space. In contrast, flow-matching models typically require more complex sampling procedures that are slower and less efficient.
>
> > (W2) The performance of the model is repeatedly attributed to the unified latent space, but no evidence is provided supporting this assertion.
> >
> > (Q3) Can you design an ablation study or other experiment that supports your claim the accuracy of your model is due to the unified latent space instead of any of the other architectural and training procedure choices you made?
>
> Thank you for this valuable suggestion. To better validate the contribution of the unified latent space, we have conducted an additional ablation experiment to compare our model with its variant which have separate latent space for equivariant (3D) and invariant (2D) modalities. Specifically, we split the unified VAE into two independent neural networks: one maintaining the R-Trans architecture and taking 2D information (atom types, bond types) as input, while the other uses a vanilla transformer structure and takes 3D information (atom types and coordinates) as input. Each network generates a separate latent space, with dimensions set to half of the original hyperparameter. The LDM is then trained on the concatenated latent space of these two networks.
>
> Table: 2D Evaluation metrics on the QM9 dataset.
> | Latent Space | FCD$\downarrow$ | AtomStable | MolStable | V&C | V&U | V&U&N | SNN | Frag | Scaf |
> |---|---|---|---|---|---|---|---|---|---|
> | Unified | 0.130 | 0.999 | 0.988 | 0.983 | 0.973 | 0.950 | 0.508 | 0.987 | 0.898 |
> | Separated | 0.351 | 0.995 | 0.952 | 0.943 | 0.920 | 0.913 | 0.341 | 0.940 | 0.682 |
>
> Table: 3D evaluation metrics on the QM9 dataset.
> | Latent Space | FCD$\downarrow$ | AtomStable | MolStable | Bond length$\downarrow$ | Bond angle$\downarrow$ | Dihedral angle$\downarrow$ |
> |---|---|---|---|---|---|---|
> | Unified | 0.881 | 0.993 | 0.935 | 7.04E-02 | 9.84E-03 | 3.47E-04 |
> | Separated | 2.356 | 0.982 | 0.872 | 18.4E-02 | 123E-03 | 7.03E-04 |
>
> These results support our claim that the unified latent space plays a key role in our improved generation quality. We attribute this improvement to the early fusion of 2D and 3D information in the VAE encoder, which allows the LDM to model a single coherent modality. In contrast, for the separated latent spaces, the LDM needs to learn over two distinct modalities and their interactions, which increases the complexity of generation.
>
> We will include this ablation and discussion in the revised manuscript.
>
> > (Q1) How do you achieve near zero reconstruction with a VAE? The latent space is variational so one would expect sampling from the generated embedding to produce different outputs. Do you only evaluate the mean? What are sampled outputs like?
>
> Thank you for the insightful question.
>
> **We evaluate the VAE posterior's sampled latents instead of mean.** Our VAE setup follows the design in Stable Diffusion [1], where we use a very small KL regularization coefficient (1e-8; see Appendix Table 11) to prioritize reconstruction fidelity over latent regularization. This results in a low-variance posterior distribution ($\mathrm{e}^{-8.1786}$), where samples from the posterior remain close to the mean. Consequently, the decoder can produce stable and high-fidelity reconstructions from sampled latents.
>
> Importantly, we do not remove the KL term entirely. While doing so still allows for accurate reconstruction, it significantly degrades the generation performance of the LDM, as shown in our ablation below. We attribute this to a key advantage of VAEs over AEs for generative modeling: the variational posterior ensures a structured and smooth latent space, which is critical for generation.
>
> Although we did not explicitly evaluate using the posterior mean as input to the LDM, we expect this to yield similarly poor results based on the observed impact of removing KL regularization.
>
> Table: Reconstruction error on the QM9 dataset.
> | KL weight | Atom Acc | Bond Acc | RMSD |
> |---|---|---|---|
> | 1E-8 | 1.0000 | 1.0000 | 0.002 |
> | 0 | 1.0000 | 1.0000 | 0.002 |
>
> Table: 2D Evaluation metrics on the QM9 dataset.
> | KL weight | AtomStable | MolStable | V&C | V&U | V&U&N |
> |---|---|---|---|---|---|
> | 1E-8 | 0.999 | 0.988 | 0.983 | 0.973 | 0.950 |
> | 0 | 0.995 | 0.961 | 0.982 | 0.962 | 0.950 |
>
> Table: 3D evaluation metrics on the QM9 dataset.
> | KL weight | AtomStable | MolStable |
> |---|---|---|
> | 1E-8 | 0.993 | 0.935 |
> | 0 | 0.982 | 0.833 |
>
> > (Q2) To what extent is this model permutation invariant?
>
> Our model is permutation equivariant, not invariant, in both the VAE and LDM components. This is because both modules produce atom-wise outputs and are implemented using Transformer-based architectures. Since the Transformer is inherently permutation equivariant—meaning that permuting the input atoms results in a corresponding permutation of the output—the overall model preserves this property.
>
> References:
>
> [1] High-Resolution Image Synthesis with Latent Diffusion Models. Rombach et al., CVPR 2022.

---

> > ### Comment · Reviewer_78B5 · 2025-08-01
> >
> > I find the additional results compelling and will likely be raising my score.
> >
> > I have no further questions, but the discussion with another reviewer about acknowledging other learned equivariance approaches made me think about how this is not a new idea (e.g., it is the approach underlying GNINA's [CNNs](https://pubs.acs.org/doi/full/10.1021/acs.jcim.6b00740)) but the early equivariance work (e.g. [Taco Cohen](https://proceedings.neurips.cc/paper_files/paper/2018/file/488e4104520c6aab692863cc1dba45af-Paper.pdf)) pretty convincingly demonstrates the advantages of a "baked-in" equivariant architecture.  While I find the empirical results of the paper convincing that learned equivariance is sufficient, perhaps a direction to point out for future work it to try to figure out how to get the best of both worlds.

---

> > > ### Author Response · Authors · 2025-08-01
> > >
> > > Thank you very much for your supportive and thoughtful response.
> > >
> > > We truly appreciate your encouraging feedback and are glad that the additional experiments helped clarify the effectiveness of our unified latent space design. We also found your suggestion to combine learning with built-in equivariance very insightful. While learning equivariance from data offers flexibility and generality, built-in equivariant architectures provide theoretical guarantees. Exploring how to combine the strengths of these two approaches is indeed a promising and relatively underexplored direction.
> > >
> > > We appreciate your acknowledgment and constructive comments and will consider this direction as part of our future work.

---

> > > ### Author Response · Authors · 2025-08-04
> > > **Follow-up on the Discussion**
> > >
> > > We sincerely appreciate your thoughtful feedback and are glad you found the additional results compelling. Regarding the discussion on baked-in vs. learned equivariance, we would like to further contribute to this valuable conversation:
> > >
> > > - **Baked-in equivariance** (e.g., 3D Steerable CNNs[2]) enforces exact symmetry constraints via specialized layers (e.g., steerable filters or Clebsch-Gordan tensor products). While theoretically rigorous, these methods often often introduce substantial architectural and computational complexity.  For example, [5] shows that achieving optimal performance can require **computational complexity between $O(L^5)$ and $O(L^6)$**, where $L$ is the spherical harmonic frequency. Although more efficient implementation exist, they often compromise expressiveness and performance.
> > > - **Learned equivariance** (e.g., AF3[3], MCF[4], and Ours) instead leverages SE(3) data augmentation during training to encourage equivariance. This allows us to retain general-purpose architectures without increasing model complexity or runtime cost. AlphaFold3 [3], in particular, demonstrates that this strategy can achieve state-of-the-art performance for protein folding.
> > >
> > > In summary, both approaches have their merits: baked-in equivariance provides theoretical guarantees, while learned equivariance favors flexibility and efficiency. We agree that exploring **hybrid models** that integrate the strengths of both paradigms is a promising research direction, and we plan to explore this avenue in future work.
> > >
> > > We also want to clarify that **the core novelty of this work lies in the design of a unified and near-lossless latent space** that eliminates the need for modality-specific diffusion architectures while preserving geometric fidelity (i.e., near-zero reconstruction error). This design simplifies the LDM's modeling process by operating on a single, coherent latent representation that integrates atom types, bonds, and 3D coordinates. This represents a major departure from prior works that rely on separate channels for each modality.
> > >
> > > While SE(3) augmentations have been adopted in earlier works [1,3,4], **they alone do not yield our unified and near-lossless latent space**. It is enabled by our carefully designed VAE encoder and the tailored reconstruction loss that balances flexibility and fidelity. See Table 7, Table 8 in Appendix, and our response to your W2 and Q1 for relevant ablation studies.
> > >
> > > Thank you again for your constructive suggestions and for taking the time to engage deeply with our work.
> > >
> > > [1] Protein−Ligand Scoring with Convolutional Neural Networks, Ragoza et al., J. Chem. Inf. Model. 2017.
> > >
> > > [2] 3D Steerable CNNs: Learning Rotationally Equivariant Features in Volumetric Data, Weiler et al., NeurIPS 2018.
> > >
> > > [3] Accurate structure prediction of biomolecular interactions with AlphaFold 3, Abramson et al., Nature 2024.
> > >
> > > [4] Swallowing the Bitter Pill: Simplified Scalable Conformer Generation, Wang et al., ICML 2024.
> > >
> > > [5] The Price of Freedom: Exploring Expressivity and Runtime Tradeoffs in Equivariant Tensor Products, Xie et al., ICML 2025.

---

### Official Review · Reviewer_uzP6 · 2025-07-03

**Clarity:** 2
**Significance:** 2
**Originality:** 2
**Rating:** 3
**Confidence:** 4

**Summary:**

In this work, the authors present a VAE-based framework that compresses the multi-modal features (positions, atom types, and bonds) of 3D molecules into a latent space. Using a diffusion model, they show that sampling on this latent space is effective and demonstrate this by conditional generation on QM9 and unconditional generation on GEOM DRUGs and the QM9 dataset. This approach, due to its disjoint training pipeline, is faster than a few existing frameworks for generation.

**Questions:**

Q1. How does the model truly ensure that the equivariance to SE(3) is truly present in the generated samples? In both the augmented and non-augmented cases.

Q2. Given that this method relies on a lossless latent space, how can to ensure that this is not overfitting on training data? Could you also report (only) uniqueness scores in Tables 2 and 5?

Q3. Could you provide additional details on how augmentation is done and added to the model?

Q4. How to update this model if one of the modalities says bonds have missing data. Would this approach overfit on the bonds present and not generate the bonds not seen before? Is this a good model for molecule discovery, and if so, could you elaborate on this?

Q5. The tsne plot is unclear to me. If this is in a Center of mass frame, how do I interpret translations? What do these plots contribute?


Other suggestions:
- Add other joint modeling papers and other missing citations in the Table for comparison.

**Ethical Concerns:**

["NO or VERY MINOR ethics concerns only"]

**Final Justification:**

The authors respond to all my questions in the rebuttal. The questions about the lack of generality for the lossless latent space still remain for me. Due to the fair response to the rest of the questions, I have updated my score for the paper.

**Limitations:**

Limitations section is not present in the main paper.

**Paper Formatting Concerns:**

- EQGAT model is not cited or referred to in the text.
- The citations listed with a '-', like 1-4, 9-11, do not have access to all the numbers.

**Quality:**

2

**Strengths And Weaknesses:**

### Strengths
1. This paper presents joint modeling of continuous and discrete features for both molecular generation and property prediction. Due to its decoupled nature of 1. training a VAE to have a latent space 2. performing diffusion on latents it is faster than a few previous methods.



### Weakness
1. The results presented in all the table do not have error bars.
2. This work only compares with a few discrete graph generation models, and leaves out a few joint modeling papers.
3. The paper misses important citations in the text/table
- PONITA, Bekkers et al: https://openreview.net/forum?id=dPHLbUqGbr
- FlowMol, Dunn et al, https://arxiv.org/abs/2404.19739
- MolFM, https://arxiv.org/pdf/2307.09484

---

> ### Author Rebuttal · Authors · 2025-07-31
>
> > (W1) The results presented in all the table do not have error bars.
>
> Thank you for the suggestion. We have now added standard deviation as error bars to the reported results in Tables 1, 2, and 3. For clarity, we summarize the results with error bars below (based on 5 independent inference runs):
>
> Table 1 (GEOM-Drugs)
> | 2D-Metric | FCD$\downarrow$ | AtomStable | MolStable | V&C | V&U | V&U&N | SNN | Frag | Scaf |
> |---|---|---|---|---|---|---|---|---|---|
> | UDM-3D | 0.692 $\pm$ 0.012 | 1.000 $\pm$ 0.001 | 0.925 $\pm$ 0.002 | 0.879 $\pm$ 0.001 | 0.913 $\pm$ 0.005 | 0.907 $\pm$ 0.004 | 0.525 $\pm$ 0.08 | 0.990 $\pm$ 0.007 | 0.540 $\pm$ 0.001 |
>
> | 3D-Metric | FCD$\downarrow$ | AtomStable | MolStable | Bond length$\downarrow$ | Bond angle$\downarrow$ | Dihedral angle$\downarrow$ |
> |---|---|---|---|---|---|---|
> | UDM-3D | 17.36 $\pm$ 0.610 | 0.852 $\pm$ 0.002 | 0.014 $\pm$ 0.000 | 9.89E-03 $\pm$ 0.02E-03 | 5.11E-03 $\pm$ 0.05E-03 | 1.74E-04 $\pm$ 0.02E-04 |
>
> Table 2 (QM9)
> | 2D-Metric | FCD$\downarrow$ | AtomStable | MolStable | V&C | V&U | V&U&N | SNN | Frag | Scaf |
> |---|---|---|---|---|---|---|---|---|---|
> | UDM-3D | 0.130 $\pm$ 0.003 | 0.999 $\pm$ 0.001 | 0.988 $\pm$ 0.002 | 0.983 $\pm$ 0.002 | 0.973 $\pm$ 0.001 | 0.950 $\pm$ 0.001 | 0.508 $\pm$ 0.05 | 0.987 $\pm$ 0.003 | 0.898 $\pm$ 0.005 |
>
> | 3D-Metric | FCD$\downarrow$ | AtomStable | MolStable | Bond length$\downarrow$ | Bond angle$\downarrow$ | Dihedral angle$\downarrow$ |
> |---|---|---|---|---|---|---|
> | UDM-3D | 0.881 $\pm$ 0.002 | 0.993 $\pm$ 0.001 | 0.935 $\pm$ 0.001 | 7.04E-02 $\pm$ 0.05E-02 | 9.84E-03 $\pm$ 0.04E-03 | 3.47E-04 $\pm$ 0.03E-04 |
>
> > (W2) This work only compares with a few discrete graph generation models, and leaves out a few joint modeling papers.
>
> We want to clarify that JODO[1], MiDi[2], and EQGAT[3] are all strong baselines with joint modeling capability.
>
> - JODO propose a joint 2D and 3D graph diffusion model that generates geometric graphs representing complete molecules with atom types, formal charges, bond information, and 3D coordinates.
> - MiDi jointly generates molecular graphs and their corresponding 3D arrangement of atoms.
> - EQGAT takes continuous atom positions, while chemical elements and bond types are categorical variables, and uses a diffusion model to generate molecules.
>
> > (W3) The paper misses important citations in the text/table
> >
> > - PONITA, Bekkers et al
> > - FlowMol, Dunn et al,
> > - MolFM,
>
> We thank the reviewer for pointing these related works, and are happy to include results for PONITA and FlowMol in the Table below for comparison. However, MolFM is not designed for the same benchmark, thus cannot be compared directly, we have included it in the related work session for comparison.
>
> | Method | Dataset | 2D-AtomStable (%) | 2D-MolStable (%) |
> |---|---|---|---|
> | FlowMol | GEOM-DRUGS | 99.0 | 67.5 |
> | Ours | GEOM-DRUGS | 100.0 | 92.5 |
> | PONITA | QM9 | 98.9 | 87.8 |
> | FlowMol | QM9 | 99.7 | 96.2 |
> | Ours | QM9 | 99.9 | 98.8 |
>
> Note that all results are copied from the original papers, and the AtomStable and MolStable reported in PONITA and FlowMol actually correspond to our 2D metrics (following the practice of EDM[4] and JODO[1]). We did not compare other evaluation metrics because they are not reported in these baselines' original paper.
>
> > (Limitations) Limitations section is not present in the main paper.
>
> The limitation section is now in Appendix due to our limited space in main paper. We appreciate the advice, and will revise to include it in the main paper in the camera-ready version.
>
> > (Formatting)
> >
> > - EQGAT model is not cited or referred to in the text.
> > - The citations listed with a '-', like 1-4, 9-11, do not have access to all the numbers.
>
> We thank the reviewer for catching this. In fact, EQGAT has been correctly cited in the Introduction section, with citation number 10, but we will also cite it in more appropriate places.
> We note that similar citation ranges are commonly used in NeurIPS papers [5,6]. That said, we agree that explicit citations improve clarity and have revised accordingly. We will correct the ambiguous dash-style citation ranges (e.g., “1-4”) to list individual references.
>
> > (Q1) How does the model truly ensure that the equivariance to SE(3) is truly present in the generated samples? In both the augmented and non-augmented cases.
>
> Thank you for the question. You can gently refer to Table 6 in our main paper. We provide RMSD (Root Mean Square Deviation) to measure equivariance in generated samples under different augmented and non-augmented scenarios. The low error validates that our latent diffusion process retains SE(3)-equivariance, as the generated samples closely match the original 3D coordinates.
>
> > (Q2) Given that this method relies on a lossless latent space, how can to ensure that this is not overfitting on training data? Could you also report (only) uniqueness scores in Tables 2 and 5?
>
> We appreciate your question. We report reconstruction error on the **test sets**, in Figure 1 (a;b) and Table 9 in Appendix. These molecules are different from those in the training dataset, eliminating the possibility that our model is overfitting the training dataset.
>
> The uniqueness scores are already reported in Tables 1 and 2, where “V&U” denotes molecules that are both valid and unique. A molecule is defined as unique only if it is also valid (following EDM[4], JODO[1])—otherwise, the notion of uniqueness becomes ill-defined. Our model achieves uniqueness scores of 0.913 on GEOM-DRUGS and 0.973 on QM9, demonstrating its ability to generate a diverse set of valid molecular structures.
>
> > (Q3) Could you provide additional details on how augmentation is done and added to the model?
>
> Yes. As described in Section 3.1 of the paper, given a molecule’s 3D coordinates $\mathbf{x}$, we apply a random rigid transformation by sampling a rotation $\mathbf{R} \sim \mathrm{SO}(3)$ and a translation $\mathbf{t} \sim \mathcal{N}(\mathbf{0}, 0.01\mathbf{I})$, and computing the augmented coordinates as $\tilde{\mathbf{x}} = \mathbf{R}\mathbf{x} + \mathbf{t}$. This transformation alters the molecule’s global orientation and position in space but preserves its internal structure. The augmented coordinates $\tilde{\mathbf{x}}$, along with other graph features (i.e., atom types and bonds), are then fed into UAE-3D, which is trained to reconstruct $\tilde{\mathbf{x}}$, thereby encouraging SE(3)-equivariance through augmentation.
>
> > (Q4) How to update this model if one of the modalities says bonds have missing data. Would this approach overfit on the bonds present and not generate the bonds not seen before? Is this a good model for molecule discovery, and if so, could you elaborate on this?
>
> We appreciate the insightful question and would like to clarify the following:
> - **Missing Bond Information can be mostly inferred using xyz2mol or OpenBabel.** Bond data can be inferred from the 3D coordinates. If we know the bonding connection is imperfect in the training dataset, we can use tools like xyz2mol and OpenBabel to pre-process the training data to verify and fill-in the missing chemical bonds. This process can handle most cases (>98%) according to the GEOM-DRUGS paper[7] These results should be good enough for a lot of cases and the little error is tolerable, given that the GEOM-DRUGs' ground truth annotation has 1.2% error of unstable molecules.
> - **Our model can function without bond information completely.** If the bond information is missing, we can train a new model without bond information as input and reconstruction target. The VAE can reconstruct the 3D coordinates without explicit bond information.
> For example, in the CrossDocked2020 dataset which contains protein-ligand complexes, the bond type information is not available for protein's amino acids, but our model can still reconstruct the 3D coordinates of the protein-ligand complexes with high fidelity as shown in the Table below:
>
> | Atom Acc  | RMSD(Å) |
> | ---  | --- |
> | 1.000  | 0.011 |
>
> > (Q5) The tsne plot is unclear to me. If this is in a Center of mass frame, how do I interpret translations? What do these plots contribute?
>
> Thank you for your comments. The t-SNE plot is intended to show the relative positions of real molecules in **latent space** (not the physical 3D coordinate system) under different coordinate transformations. Before these transformations, the molecules remain in a center of mass frame, but after the translations, they no longer remain centered.
>
> The key insight of this plot is that the latent code of UAE-3D exhibits structural changes with the real geometric motion. This illustrates the meaningful structural changes in the molecular representation in SE(3)-equivariant augmentation.
>
> References:
>
> [1] Learning Joint 2-D and 3-D Graph Diffusion Models for Complete Molecule Generation, Han et al., TNNLS 2024.
>
> [2] MiDi: Mixed Graph and 3D Denoising Diffusion for Molecule Generation, Vignac et al., ECML 2024.
>
> [3] Navigating the Design Space of Equivariant Diffusion-Based Generative Models for De Novo 3D Molecule Generation, Le et al., ICLR 2024.
>
> [4] Equivariant Diffusion for Molecule Generation in 3D, Hoogeboom et al., ICML 2022.
>
> [5] Convolutional Differentiable Logic Gate Networks, Petersen et al., NeurIPS 2024.
>
> [6] CSPG: Crossing Sparse Proximity Graphs for Approximate Nearest Neighbor Search, Yang et al., NeurIPS 2024.
>
> [7] GEOM, energy-annotated molecular conformations for property prediction and molecular generation, Axelrod et al., Scientific Data 2022.

---

> > ### Comment · Reviewer_uzP6 · 2025-08-04
> >
> > I thank the authors for responding to my questions in detail and clarifying a few points for me. I will discuss with other reviewers and make the final decision on my score.

---

> > > ### Author Response · Authors · 2025-08-05
> > >
> > > Thank you for taking the time to read our rebuttal and for acknowledging our detailed responses to your questions. We are glad that our clarifications were helpful in addressing your concerns.
> > >
> > > May we kindly ask whether our responses have sufficiently addressed the issues you raised, or if there remain any specific points that you feel still require further clarification? We would be very happy to elaborate on any aspect of our work that is still unclear.
> > >
> > > We would greatly appreciate it if you might reconsider your evaluation in light of the additional explanations, and comparisons we have provided. Your feedback is highly valuable to us, and we are committed to ensuring that all your concerns are fully resolved.

---

> ### Author Response · Authors · 2025-08-04
> **Kind Invitation for Further Discussion**
>
> Thank you again for your detailed and constructive review. We deeply appreciate the time and expertise you invested in evaluating our work.
>
> We would like to kindly invite you to take a look at our rebuttal, in which we have addressed each of your comments thoroughly. In particular:
>
> - We have **added standard deviation/error bars** to all the major result tables, as you suggested, based on 5 inference runs.
> - We have clarified that our method already compares with recent **joint modeling baselines** (JODO, MiDi, EQGAT), and we have further added **comparisons with other recent baselines** such as PONITA and FlowMol wherever applicable.
> - We have elaborated on how our model handles **SE(3)-equivariance**, **near-lossless latent space**, the **data augmentation strategy**, and **cases with missing bond information** in detail.
> - We also explained the design and intention behind the **t-SNE plot** in latent space.
>
> We would be very grateful if you could share any follow-up thoughts or questions you may have—we are happy to engage further in discussion and clarify anything that remains unclear.
>
> Thank you again for your valuable time and feedback!

---

### Official Review · Reviewer_cDUj · 2025-07-03

**Clarity:** 3
**Significance:** 3
**Originality:** 3
**Rating:** 4
**Confidence:** 3

**Summary:**

This paper tackles the challenge of multi-modal 3D molecular generation by proposing UAE-3D, a unified variational autoencoder capable of compressing atom types, bond types, and 3D coordinates into a single latent space while maintaining near lossless reconstruction. By leveraging SE(3)-equivariant data augmentations in training, the method learns equivariant representations without relying on 3D-specific architectural biases. The authors demonstrate that their unified latent space allows for the application of latent diffusion models (specifically, Diffusion Transformer, DiT) to generate realistic molecules efficiently and accurately.

**Questions:**

- Could the authors provide more formal justification (or empirical evidence across much more diverse datasets) for the claim that SE(3)-equivariant augmentation paired with the unified VAE suffices to ensure geometric and chemical preservation, as opposed to specialized architectures?
- What is the effect of varying the latent space dimension, depth of R-Trans encoder, DiT architecture (vs other transformer-based or non-transformer LDM backbones), or hyperparameters $\gamma$, $\lambda$? Is the model robust to such design choices or sensitive in particular to one?
- Are there patterns to the errors or invalid/implausible generations outside of what is summarized quantitatively in Tables 1/2? Could the authors expand with more systematic failure analysis (qualitative or quantitative)?

**Ethical Concerns:**

["NO or VERY MINOR ethics concerns only"]

**Final Justification:**

I will keep the rating unchanged

**Limitations:**

The limitations section is in the appendix, but should be summarized more explicitly in the main paper.

**Paper Formatting Concerns:**

-

**Quality:**

3

**Strengths And Weaknesses:**

**Strengths**
- The paper is methodologically well-grounded. The approach replaces separate modal latent spaces for molecular generation with a unified latent space, validated by strong empirical results. The SE(3)-equivariant augmentations for learning equivariant representations are elegantly simple and effective, and the use of a Relational Transformer allows seamless integration of node and edge features.

- As visualized in Figure 2 , the model achieves 100% atom/bond accuracy and extremely low coordinate RMSD on reconstruction—substantially better than prior work (GeoLDM) and critical for downstream generative modeling. Demonstrates that training and inference times are significantly reduced compared to strong baselines (GeoLDM and JODO), largely due to unified latent modeling and use of a general diffusion backbone.

- The model and pipeline architecture are clearly depicted in Figure 3, providing an accessible overview of feature extraction, latent encoding, and generative modeling.

- This paper offers insightful ablations, showing the value of each key component (unified latent space, DiT backbone, data augmentation) and the effects on RMSD and geometry metrics. Figures 4(a-c) (t-SNE plots) demonstrate latent representation consistency under SE(3) transformations, aligning with the claim of geometric structure preservation.

**Weakness**
- The paper relies heavily on empirical demonstrations, lacking formal theoretical analysis of why the unified latent space retains all necessary information for 3D generation, or convergence guarantees for the learning of equivariance solely via data augmentation. While the gains are empirically solid, more discussion or theoretical intuition would improve the scientific contribution.

- The ablation studies in Section 4.5, Table 5, and Table 6 are thoughtful but could be further strengthened. For example, the effect of latent dimension size, number of R-Trans layers, or replacing the DiT backbone with other modern diffusion architectures could be explored to validate robustness and architectural choices beyond what is already discussed. It would also be instructive to see whether the benefits of unified latent space and data augmentation persist across substantially larger or more complex molecules.

- The paper claims broad applicability of the unified latent approach to diverse 3D molecular and multimodal datasets, but empirical validation is limited to QM9 and GEOM-Drugs. Explicit discussion of the limitations in generalization, scalability to much larger chemical spaces, or application to proteins/RNAs is somewhat deferred to future work.

---

> ### Author Rebuttal · Authors · 2025-07-31
>
> > (W1) The paper relies heavily on empirical demonstrations, lacking formal theoretical analysis of why the unified latent space retains all necessary information for 3D generation, or convergence guarantees for the learning of equivariance solely via data augmentation. While the gains are empirically solid, more discussion or theoretical intuition would improve the scientific contribution.
>
> Thank you for the constructive feedback. 3D molecular generation is based on data consisting of atomic types, chemical bonds, and 3D coordinates. Our method integrates this information into a unified latent space, allowing the model to learn all necessary geometric and chemical features in one space. While we currently lack rigorous theoretical analysis, we provide the following intuitive explanations:
> 1. **Near-lossless Latent Space**: The unified latent space is designed to be near-lossless, meaning it can perfectly reconstruct the original 3D coordinates and chemical information. As shown in Figure 2, our VAE achieves 100% atom/bond accuracy and near-zero coordinate RMSD on the test set. This ensures that all necessary information is preserved during encoding and decoding.
> 2. **SE(3)-Equivariant Augmentation**: By applying SE(3) augmentations during training, the model learns to be equivariant to rotations and translations. This means that the model can generate 3D coordinates that are consistent with the original molecular structure. To validate this, we conducted experiments on 1000 molecules in the test set, applying random SE(3) transformations (rotations and/or translations) and measuring the RMSD of the molecules reconstructed by the model. As shown in the following Table, The results show that the model maintains low RMSD under different SE(3) transformations, confirming its SE(3) equivariance.
>
> | Transformation | RMSD (Å) |
> |---|---|
> | No transform. | 0.002 |
> | Random rot. | 0.002 |
> | Random trans. | 0.002 |
> | Random rot. & trans. | 0.002 |
>
> > (W2) The ablation studies in Section 4.5, Table 5, and Table 6 are thoughtful but could be further strengthened. For example, the effect of latent dimension size, number of R-Trans layers, or replacing the DiT backbone with other modern diffusion architectures could be explored to validate robustness and architectural choices beyond what is already discussed.
> >
> > (Q2) What is the effect of varying the latent space dimension, depth of R-Trans encoder, DiT architecture (vs other transformer-based or non-transformer LDM backbones), or hyperparameters? Is the model robust to such design choices or sensitive in particular to one?
>
> We thank the reviewer for the insightful questions. We provide the following studies and new results:
>
> 1. **Latent Dimension Ablation**
>
>     As already shown in Appendix Table 7, we observe that too low a latent dimension fails to capture the necessary information, leading to poor performance. Increasing the dimension to 16 significantly improves performance, while further increasing it to 32 does not provide significant additional improvement. And in the updated Table 7 below, we also provide the reconstruction error for QM9 dataset for more clarity:
>
>     | Latent Dim | Atom Acc | Bond Acc | RMSD | AtomStable | MolStable | V\&U\&N | Bond Len$\downarrow$ | Angle$\downarrow$ | Dihedral$\downarrow$ |
>     | --- | --- | --- | --- | --- | --- | --- | --- | --- | --- |
>     | 4 | 0.9201 | 0.8932 | 0.080 | 0.914 | 0.325 | 0.566 | 3.27E-01 | 2.20E-01 | 4.19E-03 |
>     | 8 | 0.9998 | 0.9746 | 0.006 | 0.987 | 0.882 | 0.923 | 1.29E-01 | 1.77E-02 | 8.14E-04 |
>     | 16 | 1.0000 | 1.0000 | 0.002 | 0.999 | 0.988 | 0.950 | 7.04E-02 | 9.84E-03 | 3.47E-04 |
>     | 32 | 1.0000 | 1.0000 | 0.003 | 0.986 | 0.867 | 0.918 | 9.98E-02 | 1.42E-02 | 7.77E-04 |
>
>     Likewise, dimensions 4 and 8 yield higher reconstruction error, indicating they are insufficient to capture molecular complexity. The latent dimension of 16 achieves near-zero reconstruction error, while 32 does not significantly improve performance and even slightly increases the error, indicating that too large a latent dimension may not be beneficial for reconstruction.
>
> 2. **R-Trans Layer Depth Ablation (VAE reconstruction)**
>
>     | R-Trans layers | Atom Acc | Bond Acc | RMSD |
>     | --- | --- | --- | --- |
>     | 3 | 1.0000 | 0.9999 | 0.00326 |
>     | 6 | 1.0000 | 1.0000 | 0.00203 |
>     | 9 | 1.0000 | 1.0000 | 0.00229 |
>     | 12 | 1.0000 | 1.0000 | 0.00209 |
>
>     We observe that decreasing the number of our transformer layers to 3 leads to a significant increase in reconstruction error, while increasing it to 9/12 can not further improve the reconstruction quality because the reconstruction error is already near-zero.
>
> 3. **Backbone Ablation**
>
>    As shown in Table 4, we already compared our DiT backbone with a standard Transformer architecture. In the updated Table 4 below, we add a comparison with PerceiverIO, a modern transformer-based architecture designed for structured inputs and outputs. The results show that DiT consistently outperforms PerceiverIO across all metrics, which we attribute to its adaptive LayerNorm layers. These layers enable DiT to effectively handle data with different noise scales, thereby improving diffusion performance.
>
>    | Structure | 3D-AtomStable | 2D-AtomStable | V\&C | V\&U | V\&U\&N |
>     |---|---|---|---|---|---|
>    | Transformer | 0.983 | 0.997 | 0.938 | 0.922 | 0.922 |
>    | PerceiverIO | 0.972 | 0.990 | 0.933 | 0.931 | 0.931 |
>    | DiT | 0.993 | 0.999 | 0.983 | 0.973 | 0.950 |
>
>    Moreover, when paired with other diffusion neural architectures, like Transformer and PerceiverIO, our model still achieves meaningful and comparable performances. The performance can also be potentially improved given more hyperparameter tuning, which is unfortunately not available given rebuttal's limited time. This observation demonstrate that our UAE-3D model is robust to different diffusion neural architectures.
>
> > (W2) It would also be instructive to see whether the benefits of unified latent space and data augmentation persist across substantially larger or more complex molecules.
> > (W3) The paper claims broad applicability of the unified latent approach to diverse 3D molecular and multimodal datasets, but empirical validation is limited to QM9 and GEOM-Drugs. Explicit discussion of the limitations in generalization, scalability to much larger chemical spaces, or application to proteins/RNAs is somewhat deferred to future work.
>
> Thank you for the suggestion. We agree that broader empirical validation is important. While QM9 and GEOM-Drugs are the standard benchmarks, we provide the following evidence to support scalability:
>
> * **GEOM-Drugs** includes molecules up to **181 atoms**, which are comparable in size to small peptides. Our method achieves **perfect reconstruction** on this dataset (see Table 9), showing robustness to large-scale molecular inputs.
> * **CrossDocked2020** is a large-scale dataset of protein-ligand complexes, containing over 100,000 protein-ligand pairs and up to over **400 atoms** per complex. We trained our VAE on molecules from this dataset, including amino acids and ligands, achieving:
>
>   | Atom Acc  | RMSD(Å) |
>   | ---  | --- |
>   | 1.000  | 0.011 |
>
> These results show that UAE-3D can scale to **larger, protein-related molecules** and maintain high fidelity. We did not test bond type prediction accuracy yet because the bond type information is not available in the dataset for protein's amino acids. Adapting to full protein/RNA generation will be addressed in future work, as discussed in our Appendix A.
>
> > (Q1) Could the authors provide more formal justification (or empirical evidence across much more diverse datasets) for the claim that SE(3)-equivariant augmentation paired with the unified VAE suffices to ensure geometric and chemical preservation, as opposed to specialized architectures?
>
> Thank you for the suggestion. We are happy to provide empirical results on more diverse datasets to support our claims. As shown in the above table, we have conducted experiments on the CrossDocked2020 dataset, which contains large protein-ligand complexes. The results indicate that our model maintains low reconstruction loss and can preserve complex geometric and chemical information even on large-scale protein-ligand molecules. We are also conducting further validation experiments on larger PDB datasets to strengthen our claims.
>
> > (Q3) Are there patterns to the errors or invalid/implausible generations outside of what is summarized quantitatively in Tables 1/2? Could the authors expand with more systematic failure analysis (qualitative or quantitative)?
>
> We appreciate the suggestion and offer the following observations:
>
> - **Error Types**: During generation, failure cases often involve incorrect **aromatic bonds**, which are particularly sensitive to small coordinate perturbations. This occurs rarely but can lead to **invalid valence assignments**. Specifically, because the interatomic distance (i.e., bond length) in the generated molecules determines the bond type, for example, the carbon-carbon bond length in the benzene ring is 1.40Å, which is between a single bond (1.54Å) and a double bond (1.34Å). Therefore, during the generation process, these bonds are likely mismatched due to small coordinate shifts, resulting in incorrect bond assignments for some carbon atoms.
> - **Quantitative Insights**: Across 10k samples, invalid molecule rate remains around 1%, and ring-distribution (Table 10) closely matches that of the training set, indicating no mode collapse.
>
> > (Limitations) The limitations section is in the appendix, but should be summarized more explicitly in the main paper.
>
> The limitation section is now in Appendix due to our limited space in main paper. We appreciate the advice, and will revise to include it in the main paper in the camera-ready version.
>
> [1] Perceiver IO: A General Architecture for Structured Inputs & Outputs, Jaegle et al., ICLR 2022.

---

> > ### Comment · Reviewer_cDUj · 2025-08-04
> >
> > Thank you for your detailed response to my review. I've considered your points. While my overall assessment (score) remains unchanged, I am very much open to further discussion with you and the other reviewers.

---

> > > ### Author Response · Authors · 2025-08-05
> > > **Thanks for the feedback**
> > >
> > > Thank you very much for acknowledging our rebuttal and efforts. We are glad that our response helped address many of your concerns. Your insightful comments and positive feedback have been invaluable in improving our work. We will incorporate the suggested refinements in the final version.

---

### Note · Authors · 2025-08-13

Dear Reviewers, ACs, SACs, and PCs,

We sincerely thank you for your time and feedback. The rebuttal process has greatly strengthened our work. Below, we outline our main contributions, the clarifications and experiments added during rebuttal, and our plans for the final version.

**Positive feedback from reviewers:**

- **[Reviewer cDUj, uzP6] Novelty & Technical Contribution:** Our work introduces a novel unified VAE framework for encoding multi-modal 3D molecular information in a nearly lossless latent space, enabling direct diffusion modeling in this space.
- **[Reviewer eGEZ] Technical Soundness:** The unified latent space and diffusion transformer design were recognized as technically sound and well-formulated.
- **[Reviewer uzP6, eGEZ] Experimental Thoroughness:** Comprehensive experiments across multiple tasks, datasets, and baselines convincingly demonstrate the advantages of our approach.
- **[Reviewer cDUj] Presentation Clarity:** Our paper presents clear motivation and an intuitive method, making the technical contributions accessible to the broader ML and computational chemistry communities.

**Key improvements during rebuttal:**

- **[Reviewer cDUj, 78B5] Robustness to architectural choices**: Added ablation studies on latent dimension, R-Trans layer depth, backbone architecture and separate latent spaces.
- **[Reviewer cDUj, uzP6] Scalability to larger molecules**: Conducted experiments on larger and more complex molecules, including protein–ligand complexes, demonstrating scalability and generalization.
- **[Reviewer uzP6, 78B5] Comparison with recent baselines**: Added comparisons with FlowMol, PONITA and SemlaFlow, showing superior performance.

All of these results and response were well received, with **no follow-up questions or concerns after our rebuttal**. Reviewers 78B5, and eGEZ gave positive post-rebuttal feedback, with 78B5 increasing their score. However, reviewer uzP6 responded briefly without leaving any further questions for discussion or changing their score. We hope that all reviewer uzP6's concerns will be addressed in the Reviewer-AC Discussions by referring to our previous, detailed responses.

**Final version commitments:**

- **Integration of all rebuttal-phase results**: Incorporate all additional experimental results, analyses, and clarifications.
- **Clearer limitations discussion**: Move the limitations section from the appendix to the main paper for better visibility.

---

### Decision · Program_Chairs · 2025-09-17

**Decision:**

Accept (poster)

**Comment:**

This paper tackles the challenging problem of unified latent encoding for 3D molecules, given their inherently multimodal nature. The proposed autoencoder uses R-Trans to encode atom types, bonds, and coordinates into a single latent space and a generic transformer for decoding, trained with well-motivated loss functions and SE(3)-equivariant augmentations. The approach achieves low reconstruction error and enables high-quality DiT-based generation. Reviewers highlighted the significance of the unified latent space, the efficiency of the framework, and the strong empirical performance. Concerns centered mainly on missing baselines and the limited scope of evaluation, but many of these were addressed convincingly in the rebuttal. Overall, I find the contribution meaningful and well-supported, and I recommend acceptance.